# Entrainment and maintenance of an internal metronome in supplementary motor area

**Jaime Cadena-Valencia[1†], Otto García-Garibay[1], Hugo Merchant[1], Mehrdad Jazayeri[2], Victor de Lafuente[1]\***

[1]Institute of Neurobiology, National Autonomous University of Mexico, Querétaro, México; [2]McGovern Institute for Brain Research, Massachusetts Institute of Technology, Cambridge, United States

**Abstract** To prepare timely motor actions, we constantly predict future events. Regularly repeating events are often perceived as a rhythm to which we can readily synchronize our movements, just as in dancing to music. However, the neuronal mechanisms underlying the capacity to encode and maintain rhythms are not understood. We trained nonhuman primates to maintain the rhythm of a visual metronome of diverse tempos and recorded neural activity in the supplementary motor area (SMA). SMA exhibited rhythmic bursts of gamma band (30–40 Hz) reflecting an internal tempo that matched the extinguished visual metronome. Moreover, gamma amplitude increased throughout the trial, providing an estimate of total elapsed time. Notably, the timing of gamma bursts and firing rate modulations allowed predicting whether monkeys were ahead or behind the correct tempo. Our results indicate that SMA uses dynamic motor plans to encode a metronome for rhythms and a stopwatch for total elapsed time.

DOI: https://doi.org/10.7554/eLife.38983.001

**\*For correspondence:**
lafuente@unam.mx

[†]These authors contributed equally to this work

**Competing interests:** The authors declare that no competing interests exist.

## Introduction

Adaptive behavior benefits from the ability to discern temporal regularities in the environment. To exploit these regularities, the brain must be able to measure time intervals between repetitive events (*Buhusi and Meck, 2005*; *de Lafuente et al., 2015*; *Confais et al., 2012*; *Leon and Shadlen, 2003*; *Grahn and Brett, 2007*; *Merchant and Lafuente, 2014*; *Merchant et al., 2015*), and use this timing information to anticipate future events (*Goel and Buonomano, 2014*; *Jazayeri and Shadlen, 2010*; *Uematsu et al., 2017*). This behavior is evident when we dance to music, which requires perceiving rhythms and generating movements in sync with them (*Levitin et al., 2018*). Nonhuman primates and other vertebrates are capable synchronizing their movements to periodic rhythms (*Merchant et al., 2013*; *Takeya et al., 2017*; *Gámez et al., 2018*), and we recently showed that monkeys can internally maintain rhythms of different tempos in the absence of overt motor actions (*García-Garibay et al., 2016*). Ample evidence indicates that cortical and subcortical motor circuits participate in behavioral tasks that require time perception and temporally precise behavioral responses (*Mita et al., 2009*; *Crowe et al., 2014*; *Bartolo et al., 2014*; *Merchant and Averbeck, 2017*; *Grahn and Brett, 2007*; *Ivry and Spencer, 2004*; *Murray et al., 2014*). Nonetheless, the neuronal mechanisms that allow motor structures to encode rhythms of different tempos, in the absence of motor commands, are not yet completely understood.

We developed a novel visual metronome task in which nonhuman primates had to observe, and then internally maintain, a temporal rhythm defined by a *left-right* alternating visual stimulus. Crucially, subjects had to track the rhythm in the absence of overt movements (*García-Garibay et al., 2016*). By uncoupling rhythm encoding and maintenance from motor actions, we aimed to identify

**eLife digest** A catchy tune on the radio, and suddenly we are tapping our foot and moving our bodies to the rhythm of the music. We can follow a beat because our motor neurons, the nerve cells that control movements, work together in circuits. During actions that require precise timing – such as dancing to a rhythm – the motor neurons within these circuits increase and decrease their activity in complex patterns.

But recent evidence shows that these motor neuron circuits also 'switch on' simply when we perceive a rhythm, even if we do not move to it. In fact, just imagining a rhythm triggers the same symphony of electrical activity in the brain. How do motor neurons generate coordinated patterns of activity without movement or even an external stimulus?

Cadena-Valencia et al. set out to answer this question by training monkeys to follow a rhythm. The animals learned to track a dot that appeared alternately on the left and right sides of a touchscreen with a regular tempo. After a few repeats, the dot disappeared. The monkeys then had to continue mentally tracking where the dot would have been. A group of neurons in a brain region called the supplementary motor area synchronized their activity with the dot. Whenever the dot was due to appear, the neurons in the area showed a burst of rapid firing. These spikes of activity, called gamma bursts, helped the motor neurons to communicate with one another within their circuits.

The gamma bursts thus acted as an internal metronome, making it easier for the monkeys to follow the rhythm. These results should be a starting point for other studies to pinpoint exactly where and how this rhythmic activity arises, and how the brain uses gamma bursts to synchronize our movements to a tempo.

DOI: https://doi.org/10.7554/eLife.38983.002

the mechanism that allows the brain to internally maintain rhythms of different tempos. While monkeys performed the task, we recorded the local field potentials (LFPs) and spiking activity of single neurons in the supplementary motor area (SMA) that has been implicated in timing and rhythm perception (*Buzsáki et al., 2012*; *Pesaran et al., 2002*). Our results show that bursts of lower gamma band activity (30–40 Hz) reflect the internally maintained tempos by a simple mechanism: the intervals defining the rhythm are encoded by the periodic onset of gamma bursts. Moreover, increasing amplitudes of gamma bursts reflected an estimate of total elapsed time (i.e. the total time since the rhythm began). Importantly, gamma bursts encoded both rhythm and the total elapsed time in the absence of sensory stimulation and overt motor activity.

## Results

### Monkeys can perceive rhythms and maintain them internally

We trained two rhesus monkeys (*M. mulatta*) to perform a visual metronome task (*Figure 1A*). While maintaining eye and hand fixation over the screen, monkeys saw a visual stimulus that appeared on one side, switched to the other, and the back to the initial location. This alternating stimulus defined three *entrainment* intervals of an isochronous rhythm. On each trial, the interval duration was pseudo-randomly chosen to be 500, 750, or 1000 ms. In this manner, animals were presented with a visual metronome whose tempo was changed on a trial-by-trial basis (*Figure 1A*).

After the third *entrainment* interval, the visual stimulus disappeared, and subjects had to maintain the rhythm internally by keeping track of the virtual position (left or right) of the stimulus as a function of elapsed time. To test the ability of subjects to maintain the rhythms, a *go-cue* at the middle of any one of up to four *maintenance* intervals instructed the subjects to reach towards the stimulus location (the *go-cue* consisted of removing the hand fixation point; the number of *maintenance* intervals was pseudo-randomly chosen; *Figure 1A*). Thus, the key parameters in the visual metronome task were (1) interval duration (500, 750, or 1000 ms), and (2) the number of *maintenance* intervals that subjects had to wait after the visual stimulus was gone.

We characterized monkeys' ability to maintain the rhythms by plotting the proportion of correct responses as a function of the elapsed time since the initiation of the first *maintenance* interval (*Figure 1B*). The behavioral results show that monkeys satisfactorily performed the task and were

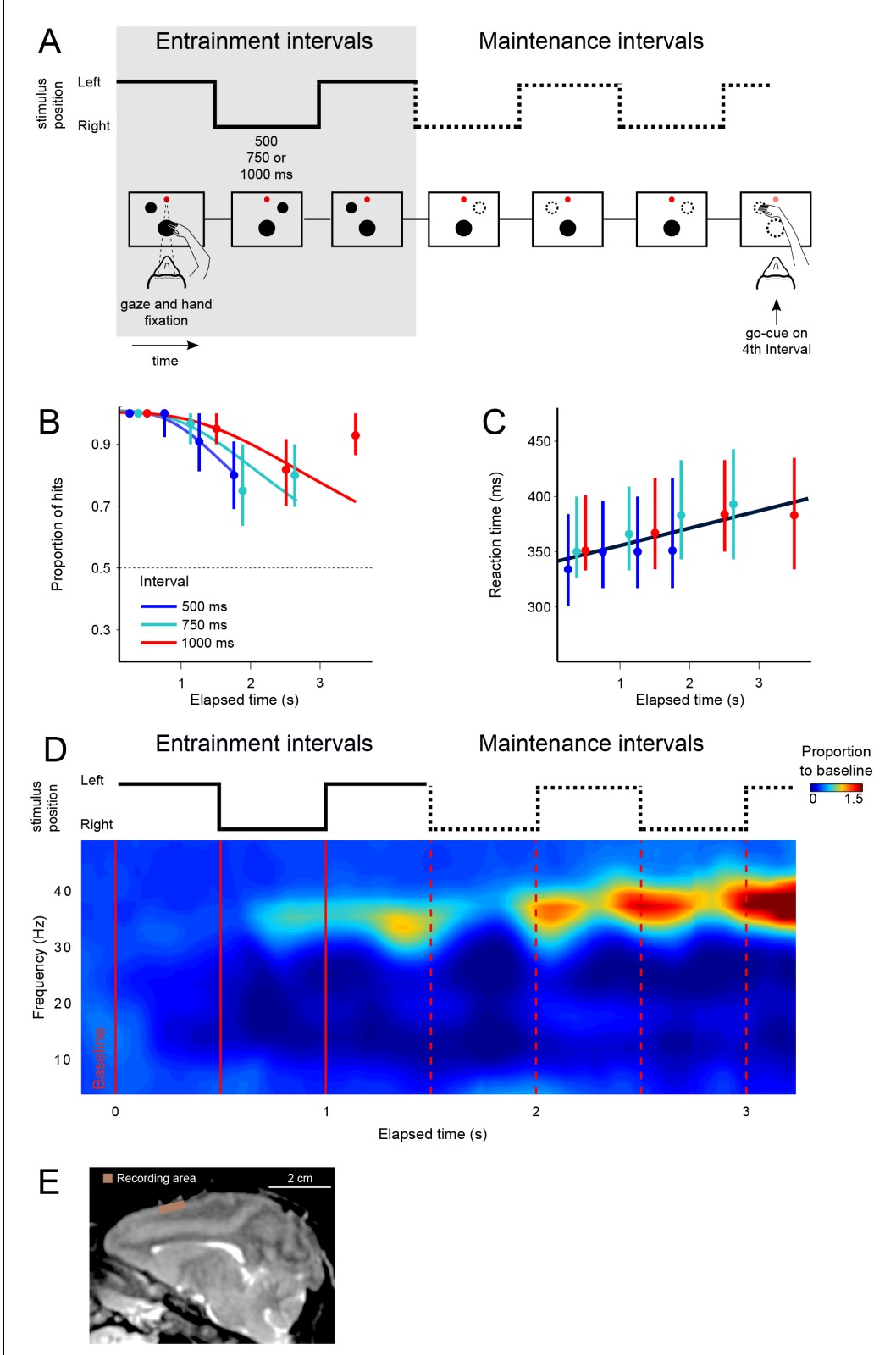

**Figure 1.** The visual metronome task. (**A**) Rhythms of different tempos were defined by a left-right alternating visual stimulus that appeared on a touch screen. While keeping eye and hand fixation, subjects first observed three isochronous *entrainment* intervals with duration of either 500, 750, or 1000 ms (pseudo-randomly selected on each trial). After the last *entrainment* interval, the visual stimulus disappeared initiating the *maintenance* intervals in which subjects had to keep track of the stimulus' virtual location (left or right, broken lines). A *go-cue* (extinction of the hand fixation) at the middle of

*Figure 1 continued on next page*

*Figure 1 continued*

any of the four *maintenance* intervals prompted the subjects to reach toward the estimated location of the stimulus. It is important to note that this was not an interception task because the left-right switching stopped at the time of the *go-cue*. Monkeys received a liquid reward when correctly indicating the stimulus location. (B) The proportion of correct responses is plotted as a function of elapsed time during the *maintenance* intervals. Colors indicate the performance for the three tempos (500, 750, 1000 ms). Performance was significantly above chance (broken line at p=0.5; z-test p<0.001; n = 131 sessions; median ±I.Q.R. over sessions). The decrease in performance as a function of elapsed time is expected from variability of the subjects' internal timing in the absence of the external visual rhythm. This drop in performance was captured by a model of timing subject to scalar variability (continuous lines). (C) Reaction times to the *go-cue* increased as a function of elapsed time (n = 131 sessions; median ±I.Q.R. over sessions). Black line indicates a linear regression on the median reaction times. (D) Mean spectrogram across recording sessions and subjects (500 ms interval). The step traces at the top indicate the stimulus position as a function of time, for *entrainment* and *maintenance* intervals. Signal amplitude was normalized with respect to a 500 ms *baseline* period before stimulus presentation. A salient modulation of the LFP signal is observed around the gamma band (30–40 Hz). Gamma activity rhythmically increases in sync with the left-right transitions of the stimulus. Note also the increase in gamma activity as a function of total elapsed time. (E) Recordings were made from the supplementary motor area (SMA). The recoding chamber on monkey 1 (shown) was centered 23 mm anterior to Ear Bar Zero and 4 mm lateral to the midline, on the left hemisphere. The image shows a sagittal plane 2 mm lateral from the middle.

DOI: https://doi.org/10.7554/eLife.38983.003

The following source data and figure supplements are available for figure 1:

**Source data 1.** Source data for the spectrograms.
DOI: https://doi.org/10.7554/eLife.38983.008
**Figure supplement 1.** Behavioral performance and LFP data for each monkey.
DOI: https://doi.org/10.7554/eLife.38983.004
**Figure supplement 1—source data 1.** Source data for the spectrogram.
DOI: https://doi.org/10.7554/eLife.38983.005
**Figure supplement 2.** Gamma amplitude and reaction times.
DOI: https://doi.org/10.7554/eLife.38983.006
**Figure supplement 2—source data 1.** Source data for the spectrogram.
DOI: https://doi.org/10.7554/eLife.38983.007

able to correctly estimate the location of the stimulus in more than 80% of trials (94 ± 0.2% monkey 1; 86 ± 0.3% monkey 2; mean ± s.e. over sessions, n = 131 sessions).

Importantly, performance as a function of time displays the hallmark of a timing task: the proportion of correct responses declines as a function of the number of *maintenance* intervals (or equivalently, elapsed time). The proportion of correct responses started close to 100% and declined to approximately 75% for the last *maintenance* intervals (last two data points for each curve). This behavior is consistent with the internal rhythm gradually drifting away from the true tempo of the stimulus (*Gibbon et al., 1997*; *Grondin, 2001*). As we described in previous work (*García-Garibay et al., 2016*), this pattern is well captured by a model in which the subject's time estimates arise from increasingly noisy (wider) distributions, described by Weber's Law of time (also called the *scalar property* of timing) (*Laje et al., 2011*). The increase in timing variability causes the subjects to eventually fall out of synchrony with the true stimulus position (getting ahead, or behind the true tempo), thus explaining the decrease in correct responses as a function of elapsed time (*Figure 1B*, the colored curves are fits of this model to the data; pooled data across monkeys; see also *Figure 1—figure supplement 1*). Behavioral performance for the 4th maintenance interval of the 750 and 1000 ms tempos is higher than would be expected, that is it is higher than the performance on the previous 3rd interval. This is likely due to the fact that our experiment only included up to four *maintenance* intervals. We speculate that monkeys exploited this information and halted the *maintenance* at the 4th interval, so that they could avoid errors due to moving onto the 5th interval. In the future, we plan to mitigate this bias by choosing the number of *entrainment* and *maintenance* intervals from an exponential distribution with a flat hazard rate.

Reaction times to the *go-cue* increases significantly in proportion to elapsed time within a narrow window ranging between 350 ms after the first maintenance interval of the fastest tempo (500 ms intervals), to 400 ms after the last interval of the slowest tempo (1000 ms intervals) (*Figure 1C*; $R^2$ = 0.72, slope = 11 ms/s, p<0.001; monkey 1 = 10.2 ms/s±0.8; monkey 2 = 11.3 ms/s ± 0.5). This increase in reaction times could be a result of the increasing difficulty in estimating the true stimulus position. As expected by scalar variability, the subject's estimate of the stimulus position becomes noisier with time, thus increasing uncertainty and the reaction time necessary to make a decision. In *Figure 1—figure supplement 2*, we provide the LFP spectrogram aligned to movement onset,

demonstrating that gamma band activity decreases, and it is replaced by low-frequency oscillations at movement onset. We also demonstrate that larger gamma band amplitudes are correlated with increased reaction times. Overall, behavioral results show that monkeys were able to entrain to a rhythm, and maintain it in the absence of sensory stimuli, and importantly, in the absence of overt motor commands.

## Gamma oscillations reveal the internally maintained rhythms

While the monkeys performed the visual metronome task, we recorded neural activity in 131 experimental sessions (84 and 47 for monkeys 1 and 2, respectively; *Figure 1E*), and analyzed the local field potentials (LFPs) within 5–80 Hz band. As a first step, we calculated the mean spectrogram for both monkeys, across all recording sessions (*Figure 1D*; 500 ms interval shown; combined data across monkeys). Modulations of LFP amplitude were especially salient in the 30–40 Hz frequencies, which we will refer to as gamma band. In this band, LFP power was up to two-fold larger than the baseline activity recorded 500 ms before trial initiation (p<0.001; permutation test of the time-frequency bins, 1000 permutations).

The LFP amplitude in the gamma band had a rhythmic structure. It increased markedly with the presentation of the last visible stimulus (3rd *entrainment* interval, *Figure 1D*), as well as near the time when the non-visible stimulus would be switching its position from one side of the screen to the other during *maintenance* intervals (*Figure 1D*; broken red lines). To test this observation quantitatively, we verified that the average gamma amplitude at the time of switches was significantly higher than halfway between them (t-test, p<0.01 for the three tempos; window sizes 1/4th of interval length; see Materials and methods). In addition to the rhythmic modulation, gamma oscillations increased in amplitude as a function of total elapsed time (*Figures 1D* and 3C; note that the last *maintenance* interval displays the largest amplitude).

The analyses so far focused on mean LFP activity across sessions. To gain further insight into the LFP dynamics supporting the maintenance of internal rhythms, we analyzed LFP amplitude modulations within single trials. The LFP recordings from single trials (band-passed at 30–40 Hz) revealed short-duration bursts during which the oscillations transiently increase in amplitude (*Figure 2B*), consistent with recent findings in the putamen (*Bartolo et al., 2014*) and the prefrontal cortex (*Lundqvist et al., 2016*). Importantly, we observed that during the *maintenance* epoch, these bursts tended to coincide with the times at which the stimulus would have changed position, as is shown by the peaks in the spectrogram of the example single trials (*Figure 2A*).

This trend is readily visualized by color-coding the amplitude of gamma oscillations and plotting all recorded trials on a single panel (*Figure 2C*). It is readily apparent that gamma bursts during *maintenance* tend to appear around the times at which the stimulus should be switching from one side of the screen to the other. This pattern is captured by the mean gamma amplitude, across trials, as a function of elapsed time (*Figure 2D*; p<0.01, t-test that compared amplitudes at the times of switch [0.5 and 1 s] versus amplitudes at the middle of the interval [0.75 and 1.25 s]; 125 ms windows).

These salient temporal features of the gamma LFP were consistent across the three interval durations (*Figure 3A*; 500, 750, and 1000 ms intervals). To better illustrate the time distribution of gamma bursts, trials were sorted according to burst-onset time in each *maintenance* interval.

It is important to emphasize that there are no motor actions during the *maintenance* intervals, and no periodic stimuli is shown on the screen. The only difference between the three groups of trials (500, 750, 1000 ms) is the tempo of the internal rhythm that subjects are maintaining. In other words, the rapid succession of the gamma bursts in the 500 ms intervals, and the more temporally distant bursts in the 1000 ms intervals, are a reflection of the subject's internal maintenance of a visuo-spatial rhythm for the fast and slow tempos, respectively. This finding reveals a neural signature of rhythms, of different tempos, that are maintained internally.

Alignment of the gamma bursts to their onset time revealed that bursts have a similar temporal profile across tempos and elapsed intervals (*Figure 3B*). Importantly, we found that the amplitude of these bursts increased in proportion to the time elapsed since the initiation of the internal rhythm (*Figure 3C*; $R^2 = 0.86$, exponential model). The results presented so far indicate that (1) the LFPs in SMA encode internal rhythms by means of gamma bursts that occur in sync with the beats (i.e. location switch) of a visual metronome presented earlier; and that (2) these bursts increase in amplitude, providing a neural correlate for total elapsed time.

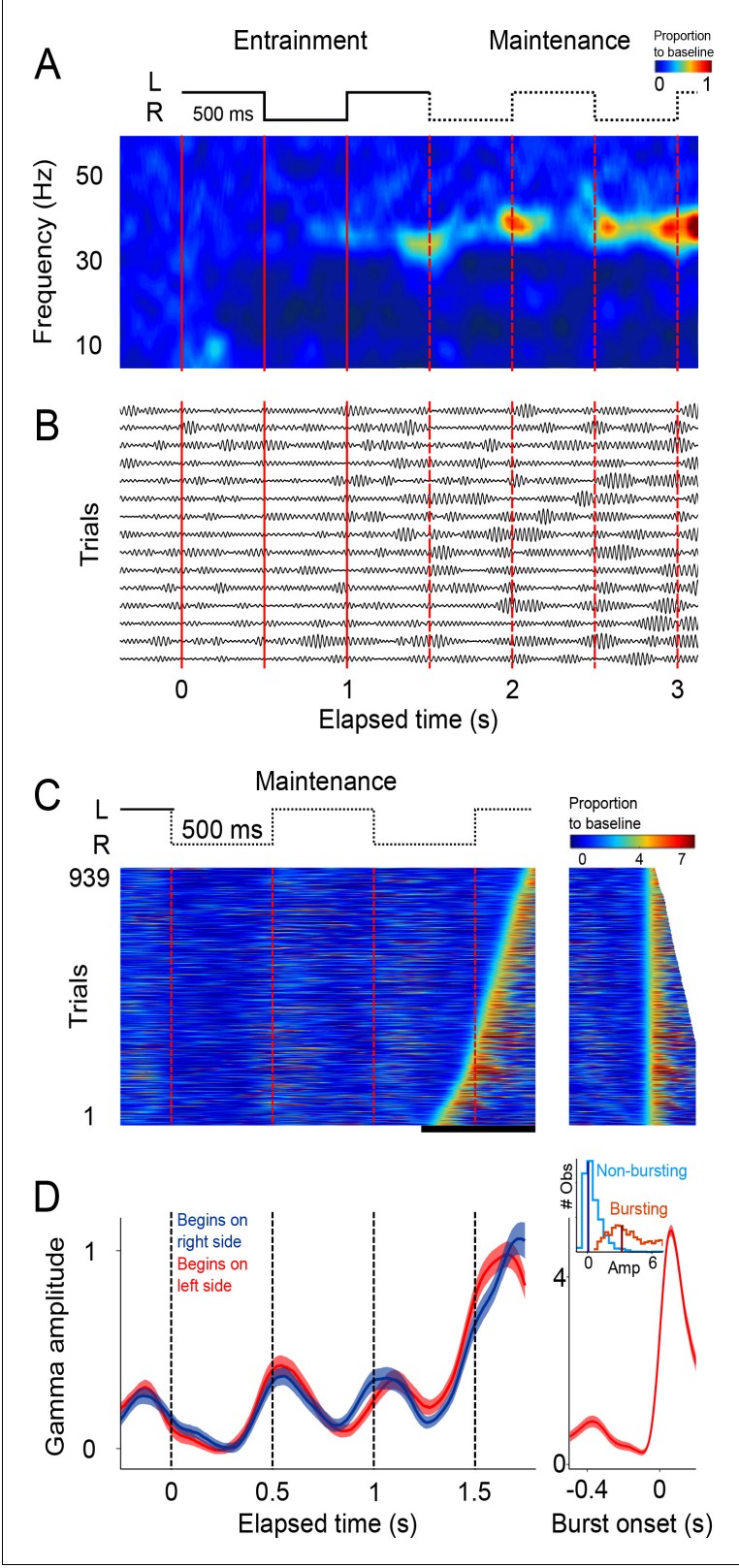

**Figure 2.** Single trial analysis of the LFP. (**A**) Representative spectrogram of 15 single trials (500 ms interval). There is an increase in amplitude at the gamma band (30–40 Hz), particularly salient during *maintenance* intervals. (**B**) Single-trials of the LFP signal, band-pass filtered at the 30–40 Hz gamma band. Gamma oscillations are composed of transient bursts during which oscillations increase in amplitude. Note how the bursts tend to occur in sync with

*Figure 2 continued on next page*

*Figure 2 continued*

left-right transitions of the stimulus and tend increase in amplitude as a function of total elapsed time. (C) Gamma amplitude on each trial is coded by color (939 trials; 500 ms interval, every trial starting on the left is shown, across sessions and subjects). Trials were sorted according to burst onset time within the window marked by the black line at the bottom. The panel on the right shows the last gamma bursts aligned to their onset time. Bursts were defined as the period in which gamma amplitude exceeded the 90th percentile of the amplitude distribution across trials, for at least 100 ms (four cycles of the gamma rhythm). (D) Mean gamma amplitude as a function of elapsed time (n = 131 sessions). Note how the periodic increases in gamma are in sync with the left-right internal rhythm during the *maintenance* intervals. The panel on the right shows the mean profile of the bursts in the last *maintenance* interval. The inset shows the distribution of the gamma amplitude during bursting (red distribution) and non-bursting (blue distribution) periods of the trials (dark vertical lines indicate the median burst amplitude for each distribution).

DOI: https://doi.org/10.7554/eLife.38983.009
The following source data is available for figure 2:

**Source data 1.** Source data for the single trial activity.
DOI: https://doi.org/10.7554/eLife.38983.010

## Errors due to deviations of the internal rhythm from the objective tempo

In a previous study, we demonstrated that human subjects tend to lag behind fast tempos and get ahead of slow ones (*García-Garibay et al., 2016*). This predicts that animals might systematically overestimate the 500 ms rhythms, and underestimate the 1000 ms rhythms. However, since animals only had two response options (left or right), it was not possible to use behavioral responses to disambiguate errors in which the animals were ahead or behind the true tempo. Nonetheless, we hypothesized that systematic over- and under-estimations of the intervals should be reflected in the patters of gamma activity in SMA. We therefore compared the profile of gamma activity on correct and error trials (*Figure 4A*). The results showed that, on fast tempo trials (500 ms interval), the dynamics of gamma on error trials was right-shifted with respect to correct trials. That is, error trials displayed slower dynamics compared to correct trials (*Figure 4A*, upper panel). This trend was captured by the power spectrums of error and correct trials, which showed that error trials indeed oscillated at lower frequencies (*Figure 4A*, inset on upper panel). Conversely, the dynamics of errors on slow tempo trials (1000 ms) resemble a left-shifted version of the correct trials, that is errors displayed faster dynamics as compared to the correct trials (*Figure 4A*; bottom panel). This pattern is captured by the power spectrums of correct and error trials, which show that error trials oscillated at higher frequencies compared to correct trials (*Figure 4A*, inset on the bottom panel). These results suggest that monkeys were lagging behind fast tempos and getting ahead of slow ones.

The internal rhythm increasingly getting out of synchrony was also demonstrated by the ability of a logistic classifier to differentiate between correct and error trials (*Figure 4B*; see Materials and methods). This analysis shows that correct and error trials are increasingly easier to classify as a function of elapsed time, just as it would be expected from a rhythm that increasingly falls out of sync with the correct tempo. This pattern holds true for a classifier that cumulatively uses gamma amplitude information as the trial develops, and also for a classifier using the information from a sliding window of constant length (*Figure 4B*). On average, monkeys tend lag behind fast rhythms and get ahead of slow ones. However, we must note that mean error activity comes from a mixture of lagging and leading tempos (see *Figure 4—figure supplement 1*). Thus, mean error activity does not necessarily reflect the half a cycle de-synchronization that must underlie incorrect responses on single trials.

## Gamma band activity in a *delayed-reach* task

Since SMA participates in the preparation of impending motor actions, it is possible that the rhythmic gamma bursts that we observed arise because this premotor area rhythmically prepare reach movements alternatively to the left and right locations of the screen. To test this possibility, we recorded the LFPs in a *delayed-reach* control task (*Hwang and Andersen, 2011*) in which subjects were required to reach to the left or the right after being cued by a briefly presented visual stimulus (*Figure 5A*). In this task, monkeys waited a pseudo-randomly chosen time (1100 to 3000 ms,

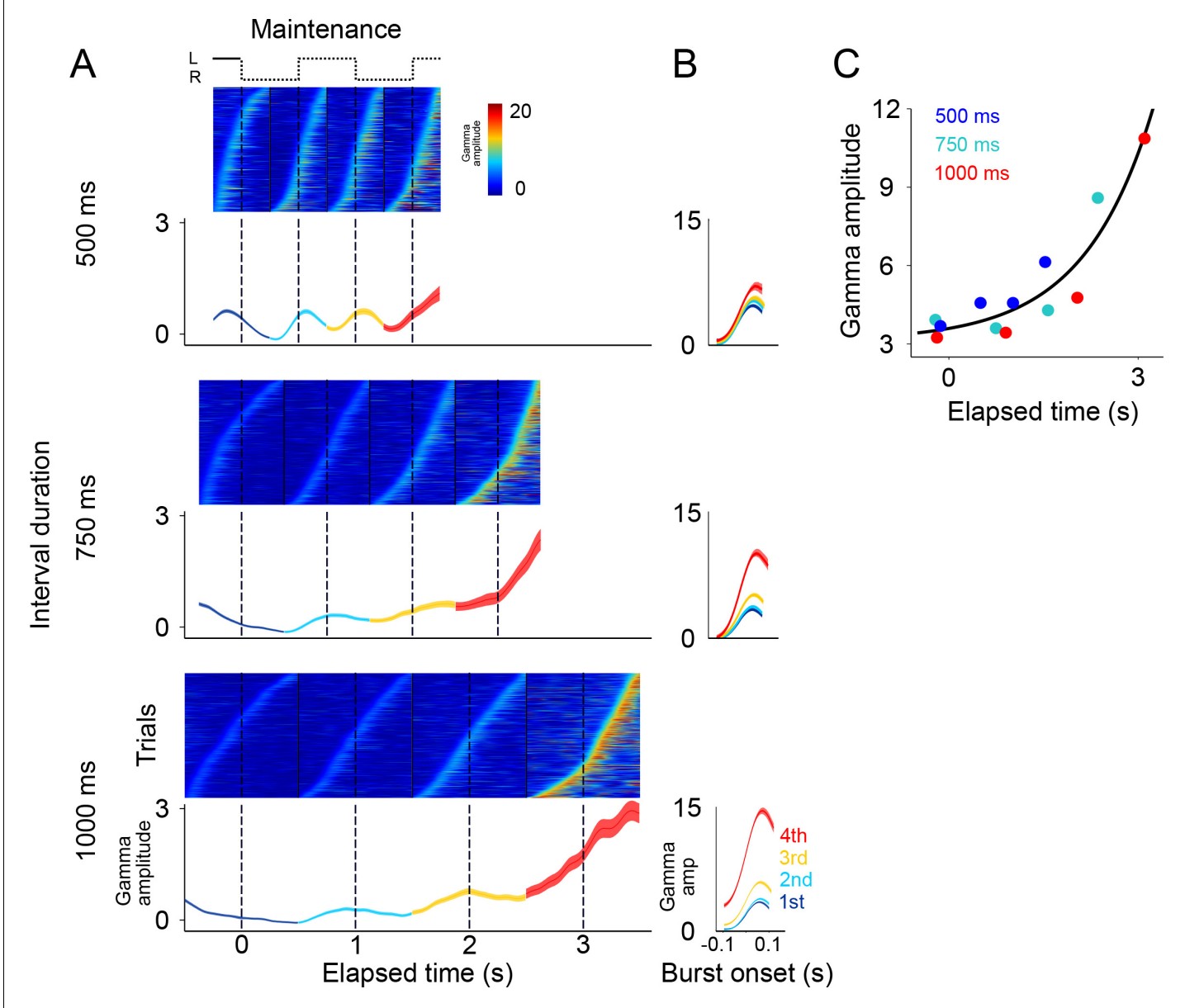

**Figure 3.** Gamma bursts in *maintenance* intervals for the three tempos (500, 750, 1000 ms). (**A**) For each *maintenance* interval and tempo, bursts are ordered according to their onset time. Below single trials, mean gamma amplitude is plotted as a function of time (trials starting on the left are shown; 131 sessions; interval duration was pseudo-randomly selected on each trial, but is grouped here for presentation; linewidth denotes s.e. across trials). (**B**) The temporal profile of gamma bursts is plotted for each stimulus transition (1st, 2nd, 3rd, and 4th, dotted lines in A), and for each interval duration (500, 750, and 1000 ms; top to bottom). The bursts have a stereotyped temporal shape and increase in amplitude after each consecutive transition. (**C**) Mean amplitude of gamma bursts plotted as a function of elapsed time for the three interval durations (500, 750, and 1000 ms; $R^2$ = 0.86, exponential model).

DOI: https://doi.org/10.7554/eLife.38983.011

The following source data is available for figure 3:

**Source data 1.** Source data for the behavioral performance.

DOI: https://doi.org/10.7554/eLife.38983.012

exponential distribution) before a go-cue prompted a reach towards the location specified by the cue (*Figure 5B*).

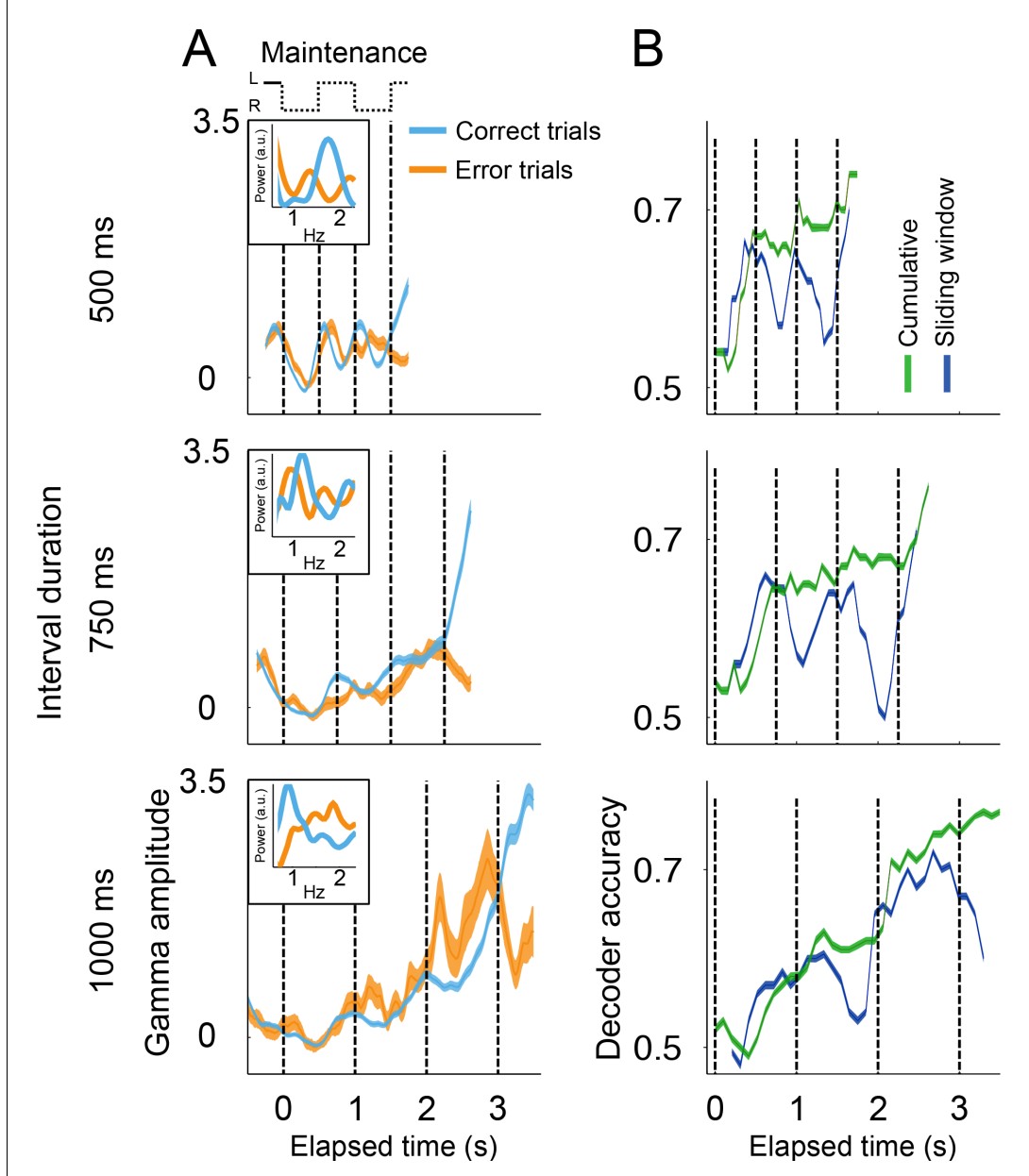

**Figure 4.** Gamma amplitude in correct and error trials. (A) Mean gamma amplitude during *maintenance* intervals, for correct (blue) and error trials (orange) (n = 1400–3000 correct, 260–540 errors; colored area shows s.e. across trials; trials starting on the left are shown, pooled across 131 sessions). The insets on each panel show the periodogram (power spectral density) of correct and error trials (mean across single trials). It can be observed that error trials oscillate at slower frequencies in the 500 ms interval trials, and oscillate at faster frequencies in the 1000 ms trials, as compared to correct responses. (B) Correct and error trials can be classified with increasing accuracy as a function of elapsed time. Two logistic classifiers were used to differentiate between correct and error trials (cross-validated on 50 correct and 50 error trials; n = 100 iterations; colored area shows s.e. across trials). One classifier used a growing window (*Cumulative*, green line) that incorporated the gamma amplitude data as each trial developed in time. The other classifier used data within a constant length window that slided across each trial (*Sliding window*, blue line).
DOI: https://doi.org/10.7554/eLife.38983.013

The following source data and figure supplement are available for figure 4:

**Source data 1.** Source data for the spectrograms.
DOI: https://doi.org/10.7554/eLife.38983.015
**Figure supplement 1.** Six-choice version of the metronome task.
DOI: https://doi.org/10.7554/eLife.38983.014

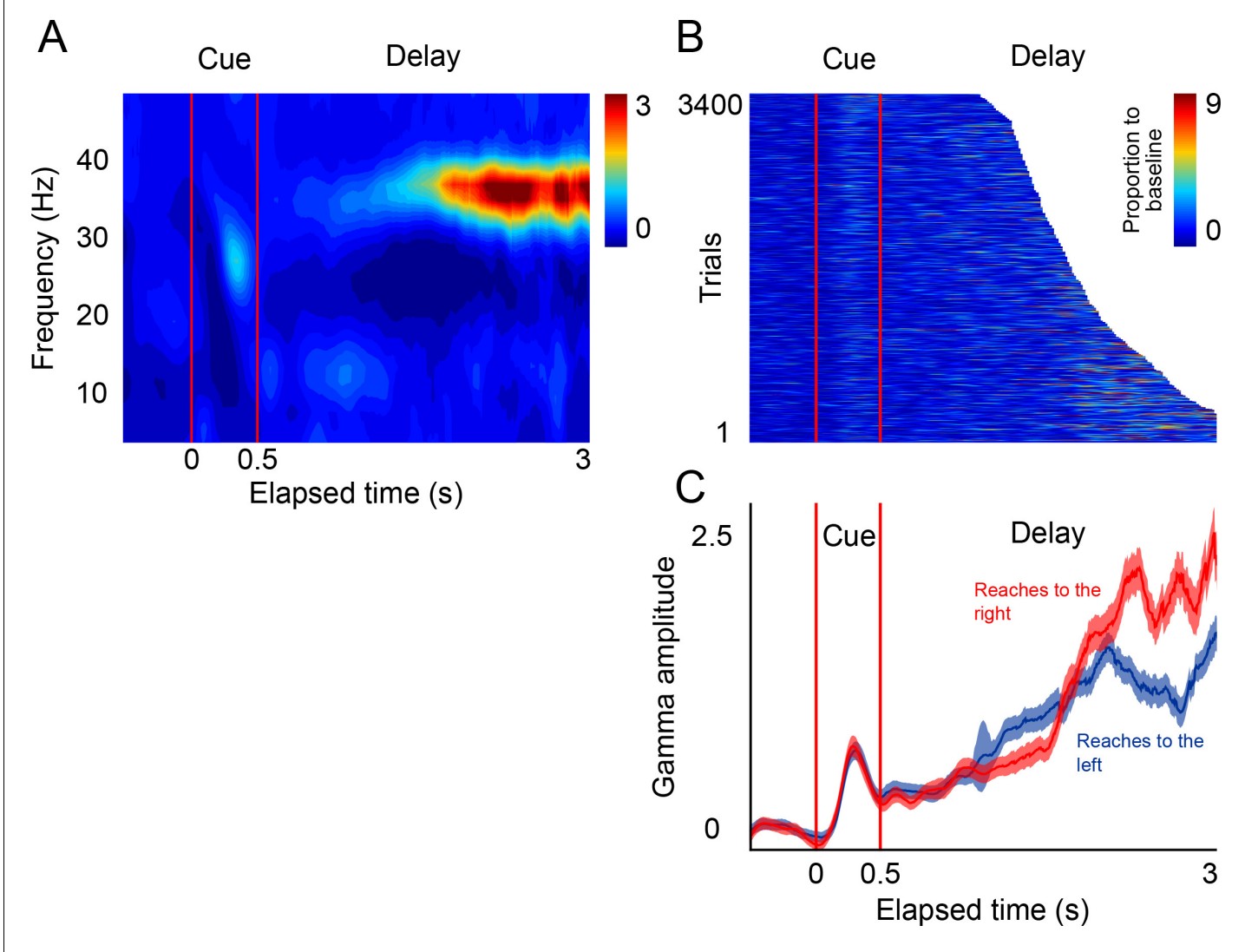

**Figure 5.** LFP activity in a *delayed-reach* task.  (A) Mean spectrogram of the LFPs during the *delayed-reach* task (reaches to the right side of the screen are shown). The stimulus presentation is indicated by the red lines a 0–0.5 s (*cue*). After the sensory cue, a variable delay between followed (1.2–3 s; exponential distribution). A salient activation of the gamma band during the delay period can be observed (n = 131 sessions). (B) Gamma amplitude across single trials of the delayed-reach task (all trials when cue was presented on right are shown). (C) Mean gamma amplitude plotted as a function of elapsed time. After a brief sensory response, gamma activity increases as a function of elapsed time. Red and blue lines indicate reaches to the right and to the left, respectively.

DOI: https://doi.org/10.7554/eLife.38983.016

The following source data is available for figure 5:

**Source data 1.** Source data for the single trial figures.
DOI: https://doi.org/10.7554/eLife.38983.017

The results of this control task show that, as monkeys prepare an impending reach movement, the LFPs in SMA generate bursts of gamma band activity that occur more frequently, and with increasing amplitude, as a function of total elapsed time (*Figure 5C*, *delay* period). These findings are consistent with the idea that gamma bursts in SMA encode impending motor commands. Moreover, the results of this control task are consistent with the idea that the SMA circuits reflect internal rhythms by means of rhythmically alternating motor plans to make a reach movement to the left and right locations of the screen.

## Gamma oscillations during entrainment of the visual metronome

According to the previous *delayed-reach* experiment, gamma bursts might be reflecting an internal rhythm by periodically alternating 'reach-left' and 'reach-right' motor plans. However, our task is designed such that a motor response was never required during the three *entrainment* intervals. For this reason, we next analyzed the gamma band activity during the *entrainment* intervals in which the presentation of the alternating visuo-spatial stimuli defined the different tempos of the visual metronome task (500, 750, and 1000 ms intervals; *Figure 6A–B*).

The results showed that even during *entrainment* intervals, which did not involve any motor planning, bursts of gamma oscillations were present in each interval, and their amplitude progressively increased after the presentation of each visual stimulus (*Figure 6A–C*). It is important to note that gamma activity in *entrainment* intervals peaked after each stimulus presentation. This is in contrast to what was observed during *maintenance* intervals, in which the peaks of gamma occurred when the stimulus switched sides. We speculate that this phase offset could be related to the process of estimating interval duration, a process that necessarily happens during *entrainment* intervals.

A potential concern is that the gamma bursts in *entrainment* intervals are merely sensory responses to visual stimuli. However, a pure sensory response should produce similar gamma dynamics after each stimulus presentation, both across consecutive entrainment intervals (1st, 2nd, 3rd), and also similar across tempos (500, 750, 1000 ms), which was clearly not the case in our results (*Figure 6A*). In particular, two observations suggest that gamma bursts during entrainment cannot be explained solely in terms of a sensory response. First, gamma bursts increased in amplitude as a function of elapsed time, but the amplitude dropped sharply 500 ms after the onset of the third *entrainment* interval (*Figures 6A,* 750 and 1000 ms panels). Therefore, gamma bursts carry information about the animals' knowledge that the third *entrainment* interval was the last visible interval, that is the last interval that could be used for estimating the tempo. Thus, gamma dynamics likely incorporate aspects of higher cognitive processing. Second, the times of burst onset do not have a fixed temporal profile with respect to stimulus presentation (*Figure 6B*). To demonstrate this, we measured the distribution of burst onset time across each consecutive interval (1st, 2nd, and 3rd *entrainment* intervals) and across metronome tempos (500, 750, and 1000 ms), and then performed Chi-squared tests between these distributions (by using burst onset time we removed the effect of burst amplitude). The tests demonstrated that the temporal profiles of gamma onset times significantly differ, both across consecutive intervals and across metronome tempos (p<0.01; corrected for multiple comparisons). In fact, gamma responses to stimulus onset are similar only during the first 500 ms of the first *entrainment* interval, which is the only epoch in which monkeys have no information about the metronome tempo (*Figure 6C*). These results indicate that gamma bursts reflect cognitive processes related to estimating the rhythm of the visual metronome.

To quantify the extent to which gamma burst amplitude encodes total elapsed time, we measured burst amplitudes in each of the three *entrainment* intervals (*Figure 6D*). We found that burst amplitude increased linearly in proportion to total elapsed time ($R^2$ = 0.94). In this manner, in addition to periodically generating bursts in each *entrainment* interval, the SMA circuit reflected the total elapsed time since the beginning of the *entrainment* epoch.

## The metronome is encoded in the firing patterns of SMA neurons

Simultaneously with LFPs, we recorded the extracellular spike potentials of 113 neurons (78 monkey 1; 35 monkey 2). The temporal profile of the mean firing rates largely resembled the modulations of gamma-band activity in the LFP in the sense that firing rates (1) display oscillatory amplitude modulations, both during *entrainment* and *maintenance* intervals (*Figure 7A,B*); (2) firing rates increase as a function of total elapsed time; and (3) the activity of neurons in the delayed-reach task increase during the period preceding reach movements to a target signaled by a brief visual cue (*Figure 7C*).

We found that SMA neurons had a preferred spatial location, that is they were more active when the stimulus was presented (entrainment intervals), or was estimated to be (maintenance intervals), on one side of the screen. Of the 113 recorded neurons, 74 preferred the right side of the screen, and 39 preferred the left side (Materials and methods). This side preference allowed us to detrend the firing rates by subtracting the mean firing rate across sides (mean between preferred and non-preferred sides of the screen; *Figure 7D*). The detrended firing rates demonstrated the cyclic

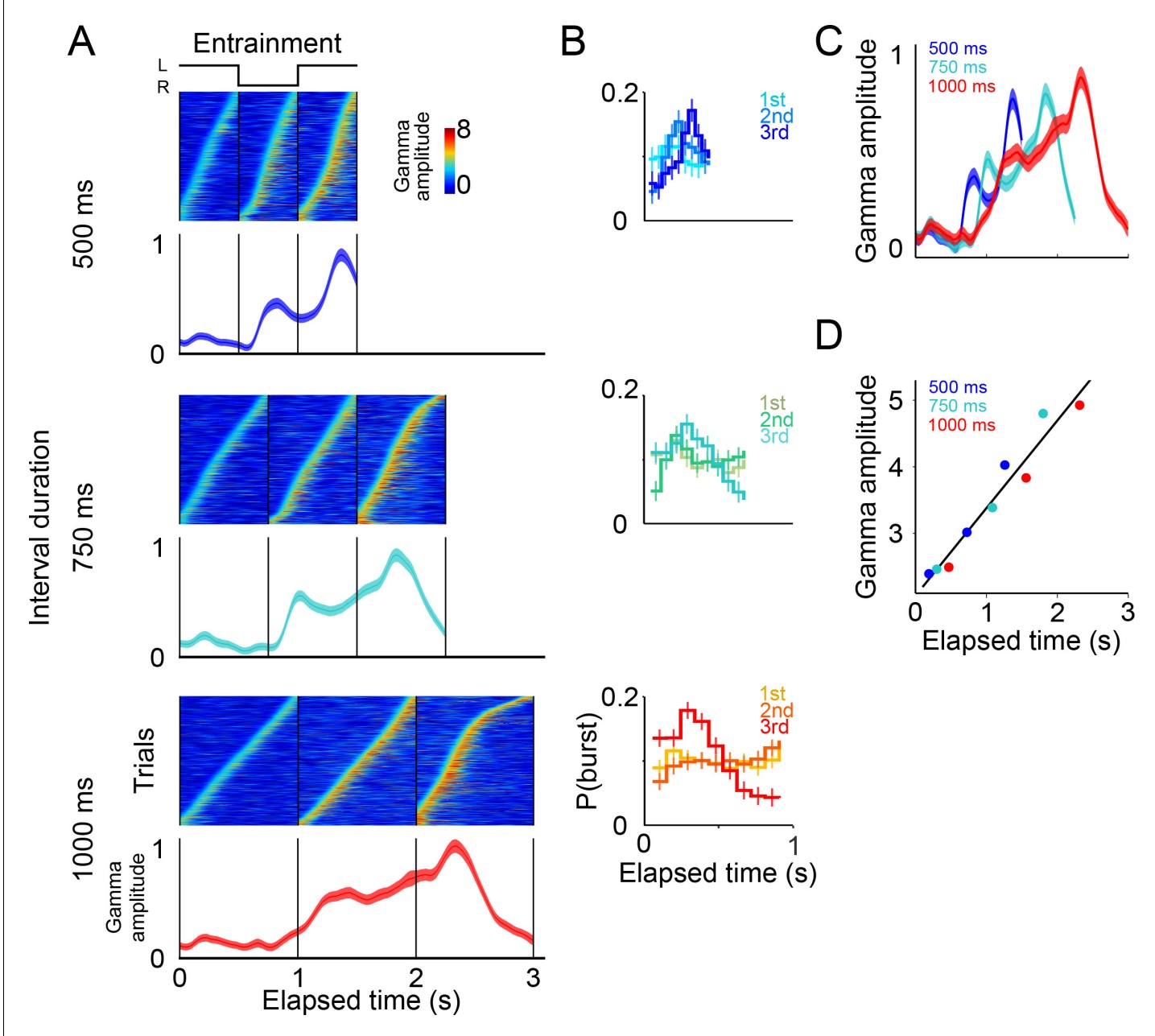

**Figure 6.** Gamma band activity in *entrainment* intervals. (A) Gamma bursts in single trials sorted by their onset time for each *entrainment* interval, and for each tempo (500, 750, and 1000 ms). Below the single-trial panels, the mean gamma amplitude is plotted as a function of time (trials starting on the left are shown; n = 131 sessions; interval duration was pseudo-randomly selected on each trial, but is grouped here for presentation; linewidth denotes s.e. across trials). (B) Probability of burst onset plotted a function of elapsed time. Line color on each panel indicates the distribution of onset times for the consecutive *entrainment* intervals (1st, 2nd, and 3rd). (C) Mean gamma dynamics for the three tempos, plotted on the same timescale (same curves as the ones below single trial panels in (A). (D) Burst amplitude as a function of elapsed time in *entrainment* intervals, for each tempo (500, 750, and 1000 ms).

DOI: https://doi.org/10.7554/eLife.38983.018

The following source data is available for figure 6:

**Source data 1.** Source data for the single trial figures.
DOI: https://doi.org/10.7554/eLife.38983.019

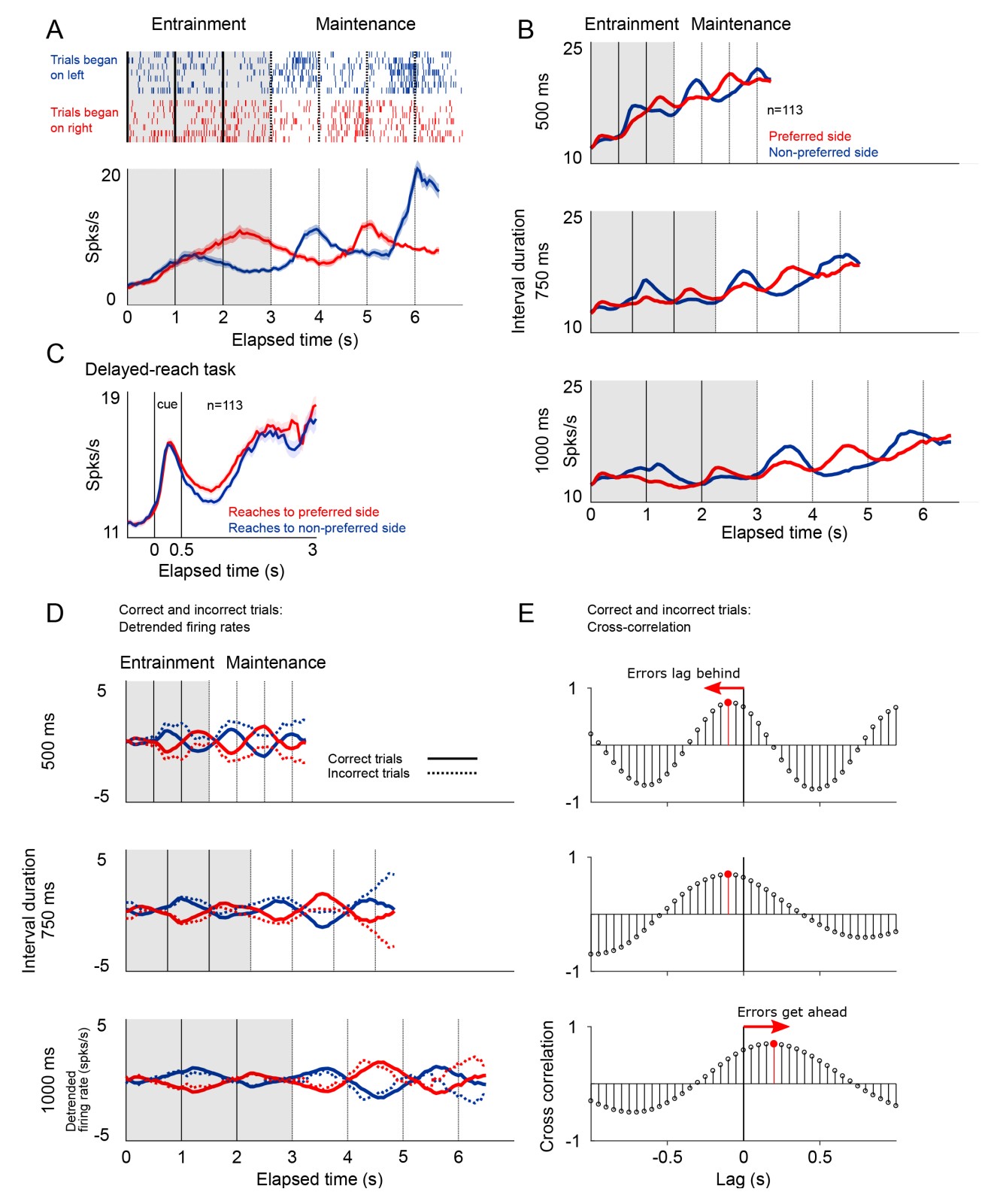

**Figure 7.** Firing patterns of SMA neurons. (**A**) Rasterplot and firing rate of a representative SMA neuron during the metronome task. This neuron fires more when the stimulus is shown (*entrainment*), or is estimated to be (*maintenance*), on the right side of the screen. The panel below shows the mean firing rates for two kind of trials. (**B**) Mean firing rate for the 113 recorded neurons show the oscillatory nature of the activity that indicates whether the stimulus is within its preferred location, or on the opposite side. Also note the increase in mean firing rates correlated with total elapsed time. (**C**) Mean

*Figure 7 continued on next page*

eLIFE Research article

Neuroscience

*Figure 7 continued*

firing rates of the 113 neurons recorded during the *delayed-reach* control task. Since the delay period had variable times (randomly selected from an exponential distribution), as time progresses fewer trial contribute to the mean (Materials and methods). (D) Detrended mean firing rates and its comparison with activity on error trials. Note how the detrending allows for a better appreciation of the oscillatory patterns of correct and incorrect trials. (E) To determine if errors were lagging behind or getting ahead of the correct tempo we calculated the cross-correlation between correct and incorrect trials for the three tempos. The upper panel, corresponding the 500 ms tempo, shows a negative lag, meaning that errors were oscillating at a slower pace as compared to correct to correct trials. The opposite effect is demonstrated on the lowest panel, corresponding to the 1000 ms tempo.

DOI: https://doi.org/10.7554/eLife.38983.020

The following source data and figure supplements are available for figure 7:

**Figure supplement 1.** Firing patterns of single neurons during the visual metronome task.

DOI: https://doi.org/10.7554/eLife.38983.021

**Figure supplement 1—source data 1.** Source data for the STA figures.

DOI: https://doi.org/10.7554/eLife.38983.022

oscillations in the activity and, importantly, allowed to calculate the cross-correlation function between correct and incorrect trials.

The cross-correlation analysis between firing rates provided an independent corroboration of the hypothesis that that errors are mostly due to the internal metronome lagging behind fast rhythms, and getting ahead of slow ones (*Figure 7E*). The cross-correlogram for the 500 and 750 ms intervals peak at negative lags, demonstrating that incorrect trials lag behind the correct tempo. The opposite pattern was observed for the slow tempo (1000 ms tempo).

## Neuronal spikes are associated with gamma band activity

To explore the relationship between single-neuron spiking and the simultaneously recorded LFP, we calculated the spike-triggered average (STA) LFP, and its spectral density, within a window of −100 to 100 ms surrounding each spike (*Figure 8A–B*, see Materials and methods) (*Denker et al., 2011*; *Fries et al., 2001*). We found that the LFP activity simultaneously recorded with each spike has a power peak at 30 Hz, and this peak is especially salient during *maintenance* intervals (*Figure 8A*, bottom panel; factorial ANOVA: interaction band/condition F = 12.11 p<0.05, Bonferroni tests of Gamma power in *maintenance* and *entrainment* vs baseline: p<0.05). Moreover, the association between spikes and the 25–40 Hz frequency band is stronger at the times of stimulus transitions, that is. around the times at which the stimulus switches from one side of the screen to the other (*Figure 8C*; window length around switch: half an interval, t-test p<0.005). To demonstrate that gamma is closely associated with the timing of spikes we performed a control analysis in which we jittered the spike times by ±15 ms with the resulting loss of the observed peak at the gamma band (random uniform distribution; grey traces *Figure 8c*; t test between jittered data in switch and non-switch conditions p=0.21).

These analyses demonstrate that the performance of the metronome task is accompanied by a tighter temporal relationship between the gamma bursts and the firing of single neurons, and this association is more prominent at the times of stimulus switching during the *maintenance* intervals. These results are consistent with previous investigations proposing that LFP oscillations near the gamma frequencies could help single neurons synchronize their firing, and thus have a larger and more temporally precise influence on downstream target structures (*Siegle et al., 2014*; *Veit et al., 2017*; *Fries, 2015*).

Finally, we demonstrate that the LFP signals we recorded reflect local interaction and were not the result of signals being volume-conducted from other brain regions. We measured the coherence between LFPs of simultaneously recorded electrodes and plotted this measure as a function of the distance between them. The results show that coherence decayed as a function of electrode distance (*Figure 8D*, $R^2 = 0.72$), as expected by an LFP signal that is generated in the neuronal circuits within the vicinity of the recording electrode.

## Discussion

Our results show that (1) monkeys can maintain rhythms in the absence of sensory stimuli and in the absence of overt motor commands. (2) Those internal rhythms are encoded by bursts of low

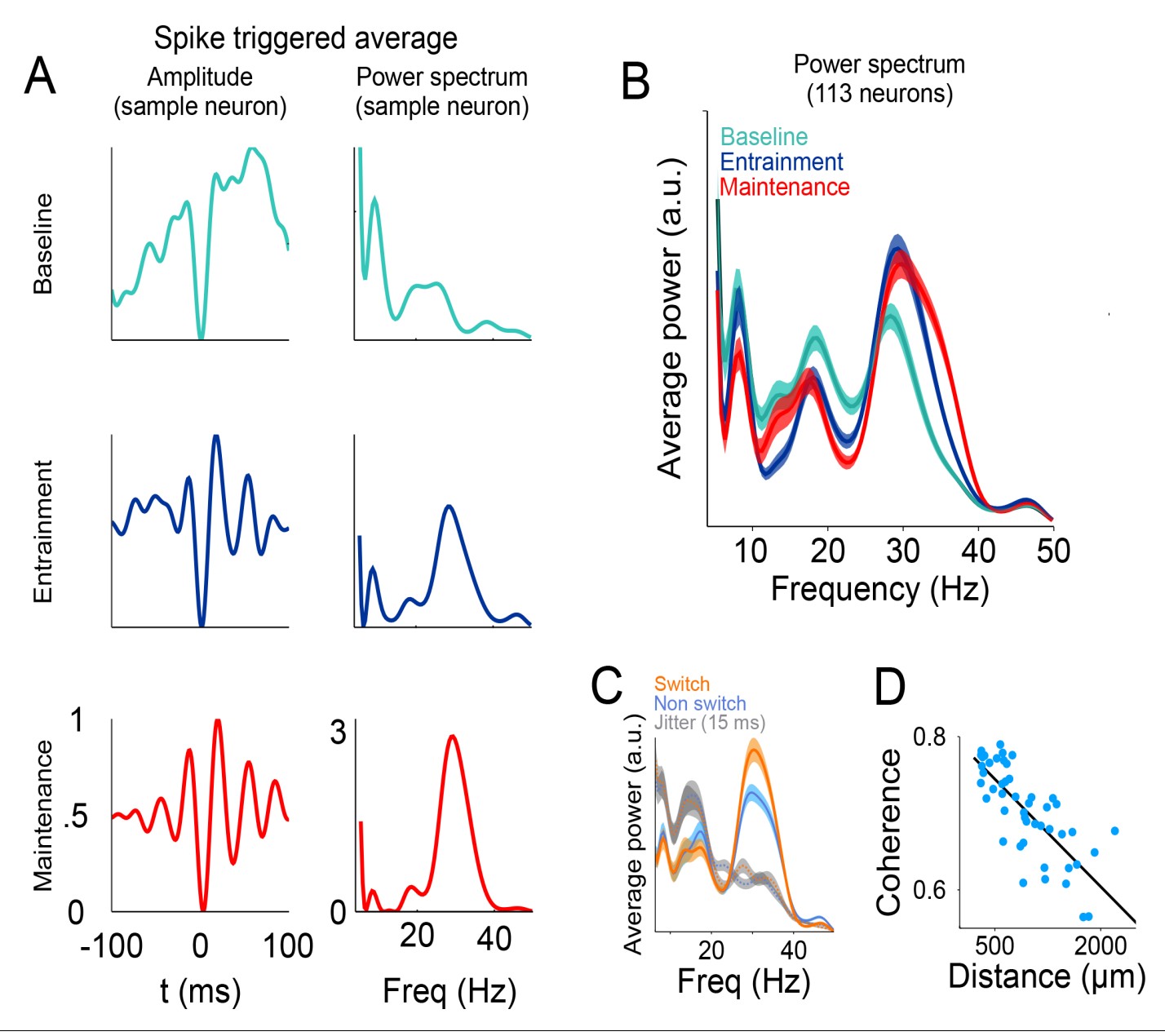

**Figure 8.** Relationship between spikes and LFP during the visual metronome task. (**A**) The three panels on the left show the spike-triggered average (STA) of the LFP signal surrounding each individual spike of an example neuron (−100 to 100 ms window centered at each spike time). The STA of three epochs is shown; *baseline* (cyan), *entrainment* (blue), and *maintenance* (red). Panels on the right show the power spectrum of the STA on each epoch. (**B**) Average STA power across neurons. Colors denote trial epochs. Note the salient power of the STA around 30 Hz (colored areas show s.e.m. across neurons; n = 113). (**C**) Average STA power for periods of stimulus switch and non-switch (windows of half the interval length, centered at times of switch or at the middle of each interval; colored areas show s.e. across neurons). Dotted lines and gray areas show the STA power obtained by jittering the spikes ± 15 ms. (**D**) Coherence between the LFPs in simultaneously recorded electrodes, as a function of distance between them. The negative slope suggests that the recorded LFPs are generated by neuronal circuits in the vicinity of the recording electrodes.

DOI: https://doi.org/10.7554/eLife.38983.023

The following source data is available for figure 8:

**Source data 1.** Source data for the STA figures.
DOI: https://doi.org/10.7554/eLife.38983.024

gamma-band LFP oscillations in SMA whose timing and amplitude indicate rhythm intervals and total elapsed time, respectively. (3) The spikes of single neurons are associated with the low gamma band frequency of the LFP, which is consistent with the idea that gamma oscillations might help to synchronize populations of neurons whose temporally coincident firing would have a larger impact on its postsynaptic targets (*Buzsáki and Schomburg, 2015*; *Cardin et al., 2009*; *Fries, 2015*; *Siegle et al., 2014*; *Veit et al., 2017*; *Womelsdorf et al., 2007*; *Wong et al., 2016*).

In our metronome task, the go-cue can arrive at the middle of any of the four *maintenance* intervals. So, there is a rhythmic modulation in the likelihood of motor response initiation. This could be related the periodic modulation of the gamma bursts and the firing rates of SMA neurons. However, we must emphasize that the rhythmicity in the gamma bursts and firing rates is also observed in *entrainment* intervals, where no movement is ever required. In addition to rhythmic modulations, the probability of the go-cue appearing, given that it has not appeared yet, increases as a function of total elapsed time (hazard rate). Thus, the increase in gamma amplitude and in the firing rates that is observed as total time elapses, could be related to the increasing likelihood of a motor response. We must note, however, that in the delayed-reach task we used an exponential distribution of delay times, resulting in a flat hazard rate. Even with a flat hazard rate, we observed increases in gamma amplitude, and in the firing rates, that are related to total elapsed time. Overall, we favor the interpretation that SMA participates in the metronome task by generating a motor plan that dynamically matches the spatio-temporal tempo defined by the rhythmic visual stimulus.

SMA plays a central role in learning, imaging, planning and executing complex motor actions (*Nachev et al., 2008*; *Romo and Schultz, 1992*; *Kurata and Wise, 1988*; *Shima and Tanji, 2000*; *Murakami et al., 2014*). It is densely and reciprocally connected to M1 and to the parietal and frontal lobes, and has direct projections to motor nuclei in the brain stem (*Jürgens, 1984*). Thanks to this diverse input and output relationships its activity been found to correlate not only with motor actions but also with cognitive, emotional, and perceptual functions (*Narayana et al., 2012*; *Vergara et al., 2016*; *de Lafuente and Romo, 2005*). SMA is active before the actual movement begins, participating in action selection, and importantly, determining the time at which actions are performed (*Merchant and Averbeck, 2017*; *Mita et al., 2009*; *Chen et al., 2010*; *Ohara et al., 2001*; *Yokoyama et al., 2016*; *Shima and Tanji, 2000*). Preparatory activity can be observed even when monkeys are required to rapidly produce a movement in response to a sensory cue (*Lara et al., 2018*), and it has been proposed that this preparatory activity constitutes the initial step of the temporal evolution of a dynamical system for the control of movement (*Churchland et al., 2010*; *Remington et al., 2018*). Consistent with this view, our results show that LFP and single neuron activity in SMA starts during the *entrainment* epoch of the metronome task, seconds before an actual movement will be required. Thus, the internal metronome is encoded as a dynamic motor plan that is initiated by the presentation of *entrainment* intervals.

That behavioral performance decreases as a function of time while gamma amplitude increases with elapsed time might seem counterintuitive. However, we must note that the tempo of the internal metronome is encoded by the timing, not the amplitude of the gamma bursts. On *Figure 4—figure supplement 1* we show a six-choice variation of the metronome task that allowed determining that error trials are not explained by random behavioral responses. Instead, the behavioral responses on the six-choice version of the task demonstrate that error trials are due to the internal metronome lagging or getting ahead of the true tempo. Even on error trials, gamma activity, and the firing pattern of neurons, show rhythmic dynamics and a mean amplitude that increases with total elapsed time. Thus, there are three independent lines of evidence supporting the notion that error trials arise from the internal metronome falling out of sync with the intended tempo. First, the periodograms of gamma activity indicate that errors on fast trials (500 ms interval) oscillate at slower frequencies as compared to correct trials. Conversely, errors on slow tempos (1000 ms) oscillate faster than correct responses. Second, these same patterns were demonstrated by the cross-correlograms of the firing rates in correct and incorrect trials (*Figure 7E*). Finally, the behavioral results on the six-choice version of the metronome task showed that errors are not uniformly distributed across choices (as would be expected from lapses of attention), but distribute around the correct stimulus position, with increasing variability for longer elapsed times, as would be expected from the scalar property of timing (*Figure 4—figure supplement 1*). Moreover, the distributions show that errors tend to be behind the true stimulus position on fast trials, and ahead on slow trials.

It has been debated whether subjects performing a rhythmic task measure individual intervals separately, or instead rely on an estimate of total elapsed time (*Laje et al., 2011*). Our results now reveal that rhythms of different tempos are supported by the presence of rhythmic neuronal activity outlining each individual interval. In addition to this, the increases in gamma burst amplitude, and mean firing rates, provide information about total elapsed time.

Gamma synchronization might be useful to the formation of local ensembles of neurons that increase the temporal coordination of presynaptic spikes on postsynaptic targets, allowing brief windows of effective communication (*Wong et al., 2016*; *Womelsdorf et al., 2007*; *Buzsáki and Schomburg, 2015*). Previous results show that gamma oscillations increase before the execution of a motor action, and then shut down at the time of movement onset (*Yokoyama et al., 2016*) a result replicated by our data. Previous work by Merchant and colleagues found that LFP gamma band activity in the basal ganglia was associated with the presentation of sensory stimuli defining the intervals within a hand tapping task (*Bartolo et al., 2014*). They found that bursts of gamma were selective for intervals of different durations, and thus different cell populations were selective for different time intervals. We found no such duration selectivity in the SMA cortex, instead observing that gamma bursts encoded intervals of different durations.

Signals associated with timing tasks can be found across multiple brain areas, including parietal, motor, and premotor cortices, as well as dopaminergic midbrain neuron in the primate (*Ghose and Maunsell, 2002*; *Genovesio et al., 2006*; *Lebedev and Wise, 2000*; *Mita et al., 2009*; *Harrington et al., 2010*). For example, *Jazayeri and Shadlen, 2015* have shown that activity of single neurons in the lateral intraparietal area encodes the time elapsed from a previous sensory stimuli, as well as the time remaining to initiate a saccadic eye movement (*Jazayeri and Shadlen, 2015*). Importantly, they showed that these signals calibrate themselves according to the underlying probability to make an eye movement within a given temporal window. A recent important result by Jazayeri and colleagues demonstrated that encoding intervals of different lengths is achieved by means of speeding up or slowing down the temporal dynamics of populations of neurons that, individually, display widely different firing patterns (*Wang et al., 2018*). Our results extend this finding to the dynamics of the LFP oscillations by demonstrating that they also show temporal scaling (*Figures 3A* and *6C*). A coherent picture is thus emerging, indicating that time-estimation and time-production signals are present as dynamic motor plans that are distributed across the motor structures that participate in executing timely motor actions.

# Materials and methods

## Subjects

Two adult male Rhesus monkeys (*Macaca mulatta*) participated in the study (weight: 5–7 kg, age: 5, 7 years). Experimental procedures were approved by the Ethics in Research Committee of the Institute of Neurobiology and were in agreement with the principles outlined in the Guide for Care and Use of Laboratory Animals (National Institutes of Health). Each monkey was surgically implanted with titanium head bolts and a titanium recording chamber over the left supplementary motor area (SMA). Placement of the chambers over the SMA was guided by structural MRI for both monkeys (*Figure 1E*).

## Behavioral task

Monkeys were trained in a visual metronome task described in detail in a previous report (*García-Garibay et al., 2016*). Briefly, while maintain eye and hand fixation over a touch screen (ELO Touch Solutions, model 1939L; ASL Eye-Track 6), subjects observed a visual stimulus (gray circle, 10° diameter, 25° eccentricity) that periodically changed position from one side of the screen to the other, at regular intervals (*entrainment* epoch; 500, 750, or 1000 ms interval; pseudo-randomly selected on each trial; *Figure 1A*). After three *entrainment* intervals the visual stimulus disappeared, and subjects had to continue estimating its position (left or right) as a function of elapsed time (*maintenance* intervals; *Figure 1A*). This visuo-spatial rhythm task is similar to a visual metronome that paces a rhythm which subjects have to keep internally during the *maintenance* epoch. To quantify the ability of the subjects to maintain rhythms of different tempos a *go-cue* (disappearance of the hand fixation area) was presented at the middle of any of the four maintenance intervals (randomly selected,

uniform distribution; *Figure 1A*). This *go-cue* instructed the subjects to make a reach movement towards the estimated target position (left or right). It is important to note that this was not an interception task, that is once the *go-cue* was presented the non-visible stimulus no longer changed position. Performance was measured as the proportion of correct responses plotted as a function of the elapsed time since the initiation of the *maintenance* epoch (*Figure 1B*). Visual stimuli and task control was achieved with the Expo software (designed by Peter Lennie, maintained by Robert Dotson; available at https://sites.google.com/a/nyu.edu/expo/).

### Delayed-reach control task

In this task, monkeys were required wait a variable delay period before making a reach movement to one side of the screen signaled by a brief visual stimulus (*Figure 5*). The stimulus appeared for 500 ms on either the left or side of the screen (randomly selected), and the delay period was randomly selected from a truncated exponential distribution with a minimum delay duration of 1.1 s and a maximum of 3 s. For analyzing the activity during the delay period (*Figures 5* and *7*), we used every trial up to the time before the go-cue. This 'attrition' method allows to use all available information up a given point in time (without the go-cue, or movement related- activity). In this manner, before 1.1 s all trials contribute to the mean activity. Then, there is a progressive attrition of trials so that for the 3 s time point ~300 trials contribute to the mean. *Figure 5B* shows every trial in which the visual cue appeared on the left.

### Neural recordings

Neural recordings were performed with seven independent movable microelectrodes (2–3 MΩ, Thomas Recordings, Giessen, Germany). Electrodes were advanced in the coronal plane into the supplementary motor area until single unit activity was obtained in at least one of the electrodes. At each recording site, spikes were isolated online (Cerebus acquisition system, Blackrock Microsystems, Salt Lake City, UT) and sampled at 30 KHz. The local field potentials (LFPs) were obtained by filtering the electrode signal at 0.5 to 500 Hz, at a 2 KHz rate. Offline, the signal was down sampled to 1 KHz, and band-pass filtered to the 2–50 Hz band.

### Data analysis

Analyses were performed with MATLAB 2013b (The Mathworks, Natick, MA), making use of the Chronux Toolbox for the time- frequency maps (*Mitra and Bokil, 2007*).

### Time-frequency decomposition

Spectral estimation was performed using multitaper methods (*Pesaran et al., 2002*; *Mitra and Pesaran, 1999*; *Cohen, 2014*). A 200 ms windows sliding at 5 ms steps was used for the time-frequency maps (one taper was used, 5 Hz bandwidth). Spectrogram power was normalized by dividing each frequency and time bin by the average power in a 500 ms *baseline* window before trial initiation.

### Oscillations of gamma amplitude

For the entrainment intervals, we compared the mean gamma amplitude (across the trials of one session) at the time of switches (dotted lines in *Figure 1D*), with the gamma amplitude in between switches (at the middle of each interval). We used window lengths of 25% the interval duration. For each tempo, each recording session contributed three pairs of switch/non-switch windows. Thus, the degrees of freedom of the t-test were (131 sessions) x (3 pairs per session) = 393–1 degrees of freedom. We performed three such tests, one for each metronome tempo (500, 750, 1000 ms). The three tests had $p < 0.01$.

### Single-trial analysis

To characterize how the amplitude of the gamma oscillations is modulated over time, we averaged the normalized spectrograms over the low gamma band frequencies (30–40 Hz). Narrow-band filtering with analytic envelopes and complex Morlet wavelet convolution yielded similar results. Gamma bursts were defined as the period of time in which gamma amplitude exceeded the 90th percentile of overall activity for at least 100 ms (i.e. for at least four cycles of the gamma oscillations). On panel

2C trials were sorted by the burst-onset time on the last maintenance interval, thus, the previous gamma bursts are not aligned. On panel 3A gamma bursts were aligned independently on each interval,that is the bursts were aligned for the first transition, then re-aligned for the second transition and so on. This was done for display purposes only; the mean gamma activity is not affected by how trials are sorted.

## Classification of correct and error trials

A logistic function was used to identify correct and error trials:

$$p(correct) = \frac{1}{1 + e^{-(\beta_0 + t_1\beta_1 + t_2\beta_2 + \dots t_n\beta_n)}}$$

where t1 correspond to the gamma amplitude in the first time-bin, t2 to the amplitude on second time bin, and so on (10 time bins per interval, 35 time-bins for each trial). Thus, the predicted behavior arises from a linear combination of the gamma activity used to fit the logistic function. The classifier accuracy was measured on 100 trials (50 correct and 50 error trials; randomly selected) not used in fitting the logistic function. Fitting and testing was repeated 100 times, randomly selecting the test trials. For the cumulative window classifier (*Figure 4B*, green line), we used the gamma amplitude on the first time-bin and then tested the accuracy of decoding, then we added the data of the second time-bin and recalculated accuracy, and so on until the last time-bin. In a second approach that we called 'sliding window', a window of 5 time-bins were used to fit the classifier and calculate accuracy. This window moved across the trial to calculate accuracy as a function of elapsed time (*Figure 4B*, blue line).

## Neuron's spatial preference

To estimate each neuron's spatial preference, we computed the cross-correlation between the stimulus position (left and right) and the mean firing rate. For this analysis, we concatenated the mean firing rate of trials starting on the left with those starting the right, and generated the stimulus position signal accordingly. The sign at the peak of the cross-correlogram tells us if increasing firing rates are significantly correlated, or anti-correlated, with the stimulus position being on the left. With each neuron's spatial preference, we were able to generate the mean firing rate of trials starting on the neuron's preferred location, and the mean firing rate of trials starting in the opposite location (*Figure 7B*). The detrended firing rates were obtained by subtracting the mean activity across all trials, from the mean firing rates of each trial type (starting on the preferred and non-preferred location; *Figure 7D*).

## Spike-triggered average (STA)

To estimate the synchronization between the spikes and the simultaneously recorded LFP, 200 ms windows centered on each spike were analyzed (*Fries et al., 2001*; *Denker et al., 2011*). The average LFP in these windows were computed and normalized peak-to-valley to values between 0 and 1. This procedure was applied before spectral decomposition of the STA (*Figure 8A*, power spectrum), allowing the comparison of spectral density maintaining the same maximum amplitude across conditions (*baseline*, *entrainment* and *maintenance* epochs). To assess statistical significance, we performed a factorial ANOVA with the factors *condition* (*baseline*, *entrainment, maintenance*), and *frequency* (alpha, beta, gamma), where the dependent variable was the average amplitude between 6 and 10 Hz for alpha, 15 to 24 Hz for beta and 30 to 40 Hz for gamma. This analysis demonstrated that the average power of the STA over the gamma band was significantly larger during *entrainment* and *maintenance*, as compared to the baseline period ($p < 0.01$; *Figure 8B*). We normalized the amplitude of the LFP traces surrounding each spike to account for the increase in gamma amplitude with total elapsed time.

## Coherence between simultaneously recorded electrodes

To assess the locality of the observed LFP oscillations we estimated the phase clustering between the LFPs in pairs of simultaneously recorded electrodes. We used the time series of all trials recorded while the monkeys performed the task. For each electrode pair, we band-pass filtered the signal (30–40 Hz) and estimated the analytic envelope to obtain the instantaneous phase. Then, for

each time point we estimated the difference angles between signals in the complex plane. The coherence was defined as the length of the average vector of all difference angles, a procedure that results in magnitudes between 1 (all difference angles are aligned to the same direction) and zero (random distribution) (*Cohen, 2014*). To quantify how coherence decreased as a function of electrode separation we grouped the distance variable into 50 bins containing the same number of observations per bin. A linear regression was then applied to these data (*Figure 8D*).

## Acknowledgements

We thank Edgar Bolaños for technical assistance and Juan Ortiz for obtaining the MRI images. Ana María Malagón helped with single neuron analysis. Ranufo Romo provided helpful comments on the manuscript. This work was supported by grants to from CONACYT Ciencia Básica 254313 (VdL), 236836 (HM); Fronteras de la Ciencia 245 (VdL), 196 (HM); and PAPIIT IN207818 (VdL), IN202317 (HM). JCV is a doctoral student from Programa de Doctorado en Ciencias Biomédicas, Universidad Nacional Autónoma de México (UNAM) and received fellowship number 486768 from Consejo Nacional de Ciencia y Tecnología (CONACYT).

## Additional information

### Funding

| Funder | Grant reference number | Author |
| --- | --- | --- |
| Consejo Nacional de Ciencia y Tecnología | Fellowship 486768 | Jaime Cadena-Valencia |
| Consejo Nacional de Ciencia y Tecnología | Fronteras de la Ciencia 196 | Hugo Merchant |
| Consejo Nacional de Ciencia y Tecnología | Ciencia Básica 236836 | Hugo Merchant |
| Universidad Nacional Autónoma de México | PAPIIT IN202317 | Hugo Merchant |
| Universidad Nacional Autónoma de México | PAPIIT IN207818 | Victor de Lafuente |
| Consejo Nacional de Ciencia y Tecnología | Ciencia Básica 254313 | Victor de Lafuente |
| Consejo Nacional de Ciencia y Tecnología | Fronteras de la Ciencia 245 | Victor de Lafuente |

The funders had no role in study design, data collection and interpretation, or the decision to submit the work for publication.

### Author contributions

Jaime Cadena-Valencia, Data curation, Software, Formal analysis, Investigation, Methodology, Writing—original draft, Writing—review and editing; Otto García-Garibay, Data curation, Investigation, Methodology, Writing—review and editing; Hugo Merchant, Funding acquisition, Methodology, Writing—review and editing, Interpretation of data; Mehrdad Jazayeri, Conceptualization, Methodology, Writing—review and editing, Data interpretation; Victor de Lafuente, Conceptualization, Supervision, Funding acquisition, Investigation, Methodology, Writing—original draft, Writing—review and editing

### Author ORCIDs

Hugo Merchant (iD) http://orcid.org/0000-0002-3488-9501
Victor de Lafuente (iD) http://orcid.org/0000-0002-1047-1354

## Ethics

Animal experimentation: Experimental procedures were approved by the Ethics in Research Committee of the Institute of Neurobiology (protocol number 046) and were in agreement with the principles outlined in the Guide for Care and Use of Laboratory Animals (National Institutes of Health).

## Decision letter and Author response

Decision letter https://doi.org/10.7554/eLife.38983.027
Author response https://doi.org/10.7554/eLife.38983.028

## Additional files

### Supplementary files

• Transparent reporting form
DOI: https://doi.org/10.7554/eLife.38983.025

### Data availability

MAT files with summary data for Figures 1-8, Figure 1—figure supplements 1 and 2, and Figure 7—figure supplement 1 have been provided. The full raw dataset is available on request to the corresponding author.

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
