## [Decision Letter]

Thank you for submitting your article "Entrainment and maintenance of an internal metronome in premotor cortex" for consideration by *eLife*. Your article has been reviewed by two peer reviewers, and the evaluation has been overseen by Richard Ivry as the Reviewing Editor and Senior Editor. The following individuals involved in review of your submission have agreed to reveal their identity: Masaki Tanaka (Reviewer #1).

The reviewers have discussed the reviews with one another and the Reviewing Editor has drafted this decision to help you prepare a revised submission.

Summary:

In this experiment, monkeys are trained to perform a task requiring the representation of rhythmic information and neural activity is recorded in SMA, asking about the role of this area in entrainment, timing, and motor preparation. An entrainment phase in which a visual stimulus jump from left to right in a periodic manner is following by a maintenance phase in which the stimulus is no longer presented and the monkey must keep an internal record of the period in order to generate a movement to the appropriate location following an imperative cue. LFP recordings in SMA show that gamma bursts occur at the extrapolated transition times, and increase in amplitude with elapsed time. A control experiment indicates that in a task that does not require entrainment, that there is also an increase in gamma over the course of a delayed response. These results indicate that SMA is sensitive to the periodic fluctuations in response likelihood, an interesting extension of prior work linking this area to motor preparation.

Overall recommendation:

In general, we had a favorable opinion of the paper, with the LFP data clearly showing rhythmic activity during the maintenance period, entrained to the preceding metronome. However, given that the behavioral data have been reported in a previous paper from your group, there was concern that the LFP data on their own do not represent a sufficient advance given interpretation issues (see below). An expanded presentation of the single unit data would really strengthen the paper. Whether this means a full presentation of that work or an expanded presentation on the most relevant issues is something you can determine.

Essential revisions:

1) Relevance of response preparation in interpreting the LFP data. You have placed a strong emphasis on the idea that the LFP data indicate that "premotor areas use dynamic motor plans to encode a metronome for rhythms and a stopwatch for total elapsed time." However, this has to be reconciled with the fact that the control experiment shows very similar changes in terms of gamma activity and clearly here there is no encoding of a rhythm or metronome. Taken together, one can say that the gamma increase is indicative of motor preparation (or temporal prediction/anticipation). This is, of course, related to elapsed time, but it is not the same as equating the activity with a metronome or a stopwatch. Similarly, the Abstract states that gamma amplitude provided an estimate of total elapsed time. This seems quite a bit stronger than what we see; the main inference to be taken from the increase in gamma power over time is that anticipation increases (given that the likelihood of the go-cue is increasing). Is it possible that gamma is in a sense encoding "certainty" or the hazard function? That is over the maintenance intervals, gamma increases because the animal is increasingly certain that a motor response will be required? We imagine that you will argue that, even if the gamma activity is a signature of response preparation/anticipation, it is rhythmic in nature across the maintenance period. That is, it rises and falls with each possible point in time at which the go-cue might appear. So the response preparation and rhythmic representation ideas are not mutually exclusive. Rather, in addition to being a signature of response preparation, SMA activity is also exhibiting sensitivity to the periodic fluctuations in response likelihood. The response preparation issue needs to be developed in the manuscript.

2) The minimal presentation of the single unit data. We understand that your plan is to report a comprehensive analysis of the single unit data in another paper, providing a brief "appetizer" here. In some ways, this was unsatisfying, especially since the single unit data have the potential to really complement the LFP data. There are places where we think the neural data would really boost the impact value of the paper. To provide two examples: 1) Your LFP analysis pools data from trials in which the animal responds left and right. Analyses of directionality at the single unit level might be useful in addressing the preparation/attention hypotheses mentioned above (at least in a specific sense, assuming the two directions are sufficiently discriminable). 2) The LFP data indicate that population gamma increases during both your experimental and control tasks, and thus, in both a rhythmic and non-rhythmic context. Is this similarity also seen in the single unit data? Or might we see different subpopulations of neurons? We imagine there are other insights to be gained from the single unit data.

3) Readers might be confused that performance became worse as time elapsed yet gamma power became greater. If gamma is simply signaling the anticipated time of the stimulus based on some oscillatory process, the negative relationship of these two variables is puzzling. You argue that the fall off in accuracy with elapsed time and increase in RTs are the result of scalar properties, namely increase in variability with increasing duration. While this may well be, it also seems possible that this could be an attentional effect. That is, as maintenance period grows, the likelihood that the animal falls off task increases. Some discussion of these issues would be helpful.

4) Related to the above is how to think about the analyses comparing correct and incorrect trials. You propose that incorrect trials are due to the failure to maintain the right tempo – that is, the internal rhythm might slow down or speed up, leading to the incorrect prediction of the stimulus position (when a phase shift has occurred). As above, an alternative here is an attentional one, namely that on incorrect trials the animal falls off task and guesses. Now the two hypotheses make different predictions. The attention hypothesis would be that the gamma oscillations should more or less disappear. The incorrect tempo hypothesis would predict that the gamma oscillations should persist, but either speed up or slow down. It may not be possible to discriminate these hypotheses with your data. First, a reduction in gamma power/signature might come about even if the tempo hypothesis is correct since you would be blending speed up and slow down trials. Second, the subtle shifts discussed (Figure 4) about 500 becoming slower and 1000 faster on incorrect trials might emerge from a regression towards the mean.

5) The results should be presented in a more consistent manner. You performed 131 recording sessions in two monkeys and combined the data from all sessions (Figures 1, 4, 5, 8) or presented the data obtained from representative sessions (Figures 2, 3, 6, 7). The variability across sessions and monkeys are reported for behavioral performance (Figure 1 and relevant text), but not for the neural data. You need to clearly explain how the data are pooled for each figure. It would also be helpful to see the data for each monkey separately in supplementary figures.

6) Figure 4A shows that the shift of periodic gamma band activity in error trials was only subtle and was not as large as half cycle of periodic stimulus. Although the most critical difference is seen at the time of go-cue, the difference in neural activity was minimal here. This indicates that the changes in gamma power may not account for the behavioral difference. You should address this point.

7) If the gamma bursts are associated with motor preparation one might ask whether there is a relationship between gamma and reaction time. For example, within a given maintenance interval was there a correlation between gamma and RT?

8) The paper doesn't provide sufficient methods concerning the control experiment. We assume that the animal gets a visual cue on one side and then waits during a variable delay for the go-cue. Assuming this is right, Figures 5A and 5C are puzzling since they don't divide trials by the duration of the delay. This means that they are pooling trials in which the go-cue has occurred and trials in which it has yet to appear. This seems problematic. Figure 5B doesn't do this, but it isn't clear what one is to take from this panel.

---

## [Author Response]

Essential revisions:1) Relevance of response preparation in interpreting the LFP data. You have placed a strong emphasis on the idea that the LFP data indicate that "premotor areas use dynamic motor plans to encode a metronome for rhythms and a stopwatch for total elapsed time." However, this has to be reconciled with the fact that the control experiment shows very similar changes in terms of gamma activity and clearly here there is no encoding of a rhythm or metronome. Taken together, one can say that the gamma increase is indicative of motor preparation (or temporal prediction/anticipation). This is, of course, related to elapsed time, but it is not the same as equating the activity with a metronome or a stopwatch. Similarly, the Abstract states that gamma amplitude provided an estimate of total elapsed time. This seems quite a bit stronger than what we see; the main inference to be taken from the increase in gamma power over time is that anticipation increases (given that the likelihood of the go-cue is increasing). Is it possible that gamma is in a sense encoding "certainty" or the hazard function? That is over the maintenance intervals, gamma increases because the animal is increasingly certain that a motor response will be required? We imagine that you will argue that, even if the gamma activity is a signature of response preparation/anticipation, it is rhythmic in nature across the maintenance period. That is, it rises and falls with each possible point in time at which the go-cue might appear. So the response preparation and rhythmic representation ideas are not mutually exclusive. Rather, in addition to being a signature of response preparation, SMA activity is also exhibiting sensitivity to the periodic fluctuations in response likelihood. The response preparation issue needs to be developed in the manuscript.

We agree with these very relevant points raised by the reviewers. We recorded from a motor-related cortical area so the patterns of activity that we see are ultimately associated with the preparation and execution of motor actions. We tried to emphasize this by stating that, in this area, the internal metronome is reflected as a “dynamic motor plan”, i.e. a motor plan that rhythmically alternates between “touch left” and “touch right” motor actions. The reviewers are correct in pointing out that, in our metronome task, the variables “total elapsed time” and “hazard rate” are correlated, so a deeper discussion on this issue is granted.

The increase in gamma amplitude that we observed as time elapsed during the maintenance epoch might be related to the increase in the probability of initiating a motor response (hazard rate). Thus, there is the possibility that the increasing gamma amplitude might be encoding movement hazard rate. However, there are some features of our results that the hazard rate-encoding hypothesis does not fully account for. It must be noted that, in the delayed-reach task, we used a variable time delay with an exponential distribution of go-cue times. This exponential distribution flattens the hazard rate during the delay period. In spite of this, the results show that gamma amplitude increases during the delay, even with a flat hazard rate. However, we want to emphasize that the metronome task does not have a flat hazard rate in *maintenance*, and the probability of the go-cue does increase with elapsed time (i.e. there is an increase in the probability that the go-cue occurs given that it has not occurred yet). For this reason, we cannot fully discard the hypothesis that, instead of total elapsed time, the increase in gamma amplitude could be coding the increasing movement hazard rate.

Having said that, it is also important to summarize the two observations that the movement hazard rate hypothesis cannot account for. (1) There is an increase in gamma amplitude during the entrainment period, in which no movement is ever required, and (2) there is an increase in gamma amplitude during the delayed-reach task even when the movement hazard rate remains constant due to the exponential distribution of delay times.

We aimed at providing a balanced discussion on these views. The new text in Discussion reads:

“In our metronome task, the go-cue can arrive at the middle of any of the four maintenance intervals. […] Overall, we favor the interpretation that SMA participates in the metronome task by generating a motor plan that dynamically matches the spatio-temporal tempo defined by the rhythmic visual stimulus.”

2) The minimal presentation of the single unit data. We understand that your plan is to report a comprehensive analysis of the single unit data in another paper, providing a brief "appetizer" here. In some ways, this was unsatisfying, especially since the single unit data have the potential to really complement the LFP data. There are places where we think the neural data would really boost the impact value of the paper. To provide two examples: 1) Your LFP analysis pools data from trials in which the animal responds left and right. Analyses of directionality at the single unit level might be useful in addressing the preparation/attention hypotheses mentioned above (at least in a specific sense, assuming the two directions are sufficiently discriminable). 2) The LFP data indicate that population gamma increases during both your experimental and control tasks, and thus, in both a rhythmic and non-rhythmic context. Is this similarity also seen in the single unit data? Or might we see different subpopulations of neurons? We imagine there are other insights to be gained from the single unit data.

We agree with the reviewers. Although we originally planned to communicate single unit results separately, the reviewers are correct in pointing out that the spiking data could complement the results of the LFP recordings. We are glad to present the results of the single unit recordings in a new multi-panel figure, and in associated texts in Results and Materials and methods sections.

Figures 7A and 7B, depict the firing rates during *entrainment* and *maintenance* epochs of sample neuron, and of the total population (n=113), respectively. Trials in which the visual metronome started on the neuron’s preferred side are depicted in red. Figure 7C shows the mean firing rate during the *delayed-reach* control task.

We are glad to communicate that the firing patterns of the recorded neurons corroborate the LPF findings in the sense that (1) they show oscillatory dynamics, both during *entrainment* and *maintenance* intervals; (2) the firing rates increase as a function of total elapsed time; and (3) the activity of neurons during the *delayed-reach* task increase during the delay period preceding the reach movement to the target signaled by a briefly presented visual cue. These results are consistent with the interpretation that SMA neurons encode a metronome by dynamically alternating between “move left” and “move right” motor plans.

The main difference between the LFP signal and the firing rate modulations, is that firing rates show a significant side preference, i.e. neurons fired more spikes when the stimulus was presented (*entrainment*), and was estimated to be (*maintenance*), on one side of the screen. We called this side the preferred spatial location (Figure 7B). Of the 113 recorded neurons, 74 showed more activity when the stimulus was on the right side of the screen (contralateral to the recording side), and 39 preferred the left side.

We also analyzed the spiking activity in the error trials. We performed a cross-correlation analysis between the mean firing rate of correct and incorrect trials (we de-trended the firing rates to keep only the oscillatory dynamics between preferred and non-preferred screen locations). We are glad to report that this analysis further supports our interpretation that error trials are mostly due to the internal metronome lagging behind fast rhythms, and getting ahead of slow ones. This is demonstrated by the cross-correlogram between correct and incorrect trials. For example, for the fast tempo, the correlogram peaks at -100 ms, showing that incorrect trials have a rightward displacement with respect to correct ones. In other words, errors are explained by the internal metronome lagging behind the correct fast tempo. The opposite pattern is observed for the slow tempo (1000 ms interval). These new results are described in the main text, Results section, and the new Figure 7. The old Figure 7, depicting sample neurons, is now a figure supplement.

The new text in Results (subsection “The metronome is encoded in the firing patterns of SMA neurons”) reads:

“Simultaneously with LFPs, we recorded the extracellular spike potentials of 113 neurons (78 monkey 1; 35 monkey 2). […] The cross-correlogram for the 500 and 750 ms intervals peak at negative lags, demonstrating that incorrect trials lag behind the correct tempo. The opposite pattern was observed for the slow tempo (1000 ms tempo).”

The new text in Methods (subsection “Neuron’s spatial preference”) reads:

“To estimate each neuron’s spatial preference, we computed the cross-correlation between the stimulus position (left and right) and the mean firing rate. […] The detrended firing rates were obtained by subtracting the mean activity across all trials, from the mean firing rates of each trial type (starting on the preferred and non-preferred location; Figure 7D).”

3) Readers might be confused that performance became worse as time elapsed yet gamma power became greater. If gamma is simply signaling the anticipated time of the stimulus based on some oscillatory process, the negative relationship of these two variables is puzzling. You argue that the fall off in accuracy with elapsed time and increase in RTs are the result of scalar properties, namely increase in variability with increasing duration. While this may well be, it also seems possible that this could be an attentional effect. That is, as maintenance period grows, the likelihood that the animal falls off task increases. Some discussion of these issues would be helpful.

We agree with the reviewers in the sense that at least some fraction of error trials could arise from a failure of attention/short-term memory to maintain the rhythm, or otherwise properly engage the task. We are confident however, that these often called “lapses” constitute a small proportion of the error trials. This assertion is backed by several observations: (1) Gamma activity shows an oscillatory pattern even on incorrect trials (Figure 4A). A failure of attention in which gamma activity loses its cyclic pattern would not have resulted in gamma showing an oscillatory pattern even on error trials. (2) The new analyses on the firing rate of single neurons support the notion that error trials are mostly due to a departure of the oscillatory frequency from the true stimulus tempo. The new Figure 7D demonstrates that, on error trials, the firing rate of neurons oscillates with an amplitude comparable with correct trials. As the new cross-correlation analysis shows (Figure 7E), it is the frequency of these oscillations that differs between error and correct responses.

The inverse relation between performance and gamma amplitude might seem counter-intuitive at first. However, we must note that gamma encodes individual intervals by the timing of the gamma bursts. Similar oscillatory patterns are observed in the firing rates of individual neurons. These observations mean that gamma amplitude and performance have a very specific relationship, i.e. it is the timing of gamma bursts that reveal the internal metronome while the amplitude of gamma signals total elapsed time (which is also related to the increasing hazard rate).

We are glad to present additional behavioral results from a 6-choice variation of the metronome task. In this version of the task the visual stimulus appeared at one of 6 positions arranged in a circular path centered on the eye fixation point. The stimulus then jumped clock or counter-clockwise (randomly chosen) defining the *entrainment* intervals, and then disappeared to initiate up to four *maintenance* intervals. The advantage of the 6-choice task is that it allowed us to measure whether the behavioral response was ahead or behind the true stimulus position, something that cannot be done with the 2-choice version. The behavioral results show that error trials are not due to random behavioral responses, as would be expected from a failure of attention. Instead, the distributions of responses are centered around the correct response position, and errors tend to lag behind the true stimulus position on fast tempos, and get ahead of the correct position on slow trials. These results further support the notion that error trials are due to the metronome increasingly getting out of the intended tempo as time elapses. These results are now presented as Figure 4—figure supplement 1. The new text Discussion relating to this issue reads:

“That behavioral performance decreases as a function of time while gamma amplitude increases with elapsed time might seem counterintuitive. […] Thus, even on error trials, gamma activity, and the firing pattern of neurons, show rhythmic dynamics and a mean amplitude that increases with total elapsed time.”

4) Related to the above is how to think about the analyses comparing correct and incorrect trials. You propose that incorrect trials are due to the failure to maintain the right tempo – that is, the internal rhythm might slow down or speed up, leading to the incorrect prediction of the stimulus position (when a phase shift has occurred). As above, an alternative here is an attentional one, namely that on incorrect trials the animal falls off task and guesses. Now the two hypotheses make different predictions. The attention hypothesis would be that the gamma oscillations should more or less disappear. The incorrect tempo hypothesis would predict that the gamma oscillations should persist, but either speed up or slow down. It may not be possible to discriminate these hypotheses with your data. First, a reduction in gamma power/signature might come about even if the tempo hypothesis is correct since you would be blending speed up and slow down trials. Second, the subtle shifts discussed (Figure 4) about 500 becoming slower and 1000 faster on incorrect trials might emerge from a regression towards the mean.

We agree that the interpretation of error trials is a critical issue. To further support our interpretation, we now provide additional behavioral results, from one of the monkeys, during a 6-choice version of the task that we describe in our response to point 3, and also in Figure 4—figure supplement 1 (18 behavioral sessions, 12,809 trials). The 6-choice task allowed us to measure whether incorrect responses were ahead or behind the true tempo. The results also demonstrate the scalar property of timing, i.e. responses become more variable with increasing total elapsed time. Importantly, the distributions of responses show that error trials on fast tempos tend to lag behind the true stimulus position and, conversely, error trials on slow tempos tend to get ahead of the true stimulus position. These results provide further support to the hypothesis that the vast majority of error trials are not due to attentional lapses. As reviewers correctly point out, attentional lapses would have resulted in the monkeys generating random behavioral responses (guesses). The results on Figure 4—figure supplement 1 show that behavioral responses are not random, but instead are distributed around the correct response, with increasing variability for longer elapsed times, as expected from the scalar property of timing.

Additionally, the new analysis of the firing rates of individual neurons provide independent support to our hypothesis that error trials are not due to lapses in attention. The new Figure 7 on the main text shows that firing rates during error trials do show an oscillatory pattern (Figure 7D). Furthermore, the cross-correlation of activity between error and correct trials shows that errors on fast trials (500 ms interval) lag behind the activity on correct trials. Conversely, the activity of error trials on slow trials (1000 ms interval) oscillates faster than on correct trials (Figure 7E).

To summarize, we now have three independent sources of information and analyses supporting the hypothesis that errors are not due to attentional lapses, but arise as the metronome falls out of sync with the true stimulus tempo: (1) The periodograms of gamma activity (Figure 4A) comparing hit and error trial dynamics, (2) the behavioral results of the 6-choice version of the task, showing the distribution of responses around the correct choice (Figure 4—figure supplement 1), and (3) the cross-correlation of the firing rates between hit and error trials (Figure 7E). Taken together, these three analyses favor the hypothesis that errors are due to the metronome failing out of sync from the true tempo. The new text in Discussion reads:

“Thus, there are three independent lines of evidence supporting the notion that error trials arise from the internal metronome falling out of sync with the intended tempo. […] Moreover, the distributions show that errors tend to be behind the true stimulus position on fast trials, and ahead on slow trials.”

5) The results should be presented in a more consistent manner. You performed 131 recording sessions in two monkeys and combined the data from all sessions (Figures 1, 4, 5, 8) or presented the data obtained from representative sessions (Figures 2, 3, 6, 7). The variability across sessions and monkeys are reported for behavioral performance (Figure 1 and relevant text), but not for the neural data. You need to clearly explain how the data are pooled for each figure. It would also be helpful to see the data for each monkey separately in supplementary figures.

We apologize for not being clear enough on how/when data was pooled across sessions and monkeys. We now state the number of sessions that went into each analysis across figures and panels. Also, as suggested, we now present behavioral and LFP data separately for each monkey on a supplementary figure (Figure 1—figure supplement 1).

– Figure legends 2C and 2D now state that they show every trial, and the average, of the 500 ms interval that started on the left, across sessions and subjects.

– Figure legend 3 now explicitly states that every trial, across sessions and subjects, is shown.

– Figure legend 4A states that data was pooled across sessions.

– Figure legends 5A and 5C state the number of sessions (n=131) and indicates that it includes all delayed-reach trials in which the cue was displayed on the left.

– Figure legend 6A now states that every trial starting on the left is shown (n=131 sessions). Linewidth denotes s.e. across trials.

– On the new Figure 7, we indicate the total number of neurons analyzed (n=113 neurons).

6) Figure 4A shows that the shift of periodic gamma band activity in error trials was only subtle and was not as large as half cycle of periodic stimulus. Although the most critical difference is seen at the time of go-cue, the difference in neural activity was minimal here. This indicates that the changes in gamma power may not account for the behavioral difference. You should address this point.

The reviewers are correct in pointing out that the gamma activity on error trials does not lag or gets ahead a complete half-cycle as compared to correct trials. For the monkeys to generate an error, the internal metronome should be ahead or behind at least half a cycle. However, we must note that the mean activity of errors comes from a mixture of trailing and leading internal tempos. Due to this mixture of trials, mean error activity might not show half a cycle lag. This mixture of error trials is well illustrated by the distributions of the behavioral responses in the 6-choice task (Figure 4—figure supplement 1). It can be seen that errors are distributed before and after the correct stimulus position. The activity in error trials do show a slowing or speeding dynamic across tempos (500 ms vs. 1000 ms) because there is a tendency or errors to be behind the true tempo in fast trials, and be ahead on slow ones. It is also important to note that the logistic classifier was able to correctly label error and correct trials in up to 70% of single trials, based on the pattern of gamma activity (Figure 4B). We now address this issue in Results as follows:

“On average, monkeys tend lag behind fast rhythms and get ahead of slow ones. […] Thus, mean error activity does not necessarily reflect the half a cycle de-synchronization that must underlie incorrect responses on single trials.”

7) If the gamma bursts are associated with motor preparation one might ask whether there is a relationship between gamma and reaction time. For example, within a given maintenance interval was there a correlation between gamma and RT?

This an interesting question. To address it we conducted a regression analysis between the amplitude of gamma and the reaction times in each trial. To avoid the general effects of gamma increasing as a function of time, and also the general effect of increasing reaction times (Figure 1C), we subtracted the mean gamma amplitude, and also the mean reaction times of each maintenance interval. Then, we performed a linear regression on the gamma and RT residuals. We found a significant positive correlation between gamma amplitude and reaction times, that is, increases in gamma are significantly associated with larger reaction times. The positive slope indicated that for each millisecond increase in reaction time, gamma increases by 0.19 in amplitude (Figure 1—figure supplement 2).

Additionally, we now provide a supplementary figure in which we plot the spectrogram of the LFP signal around the time of response movement onset. The figure shows that gamma amplitude decreases sharply before movement initiation. This result is consistent with the positive correlation between gamma amplitude and reaction time: large values of gamma mean that monkeys have not yet initiated the response movement. We address this issue in Results as follows:

“In Figure 1—figure supplement 2, we provide the LFP spectrogram aligned to movement onset, demonstrating that gamma band activity decreases, and it is replaced by low frequency oscillations at movement onset. We also demonstrate that larger gamma band amplitudes are correlated with increased reaction times.”

8) The paper doesn't provide sufficient methods concerning the control experiment. We assume that the animal gets a visual cue on one side and then waits during a variable delay for the go-cue. Assuming this is right, Figures 5A and 5C are puzzling since they don't divide trials by the duration of the delay. This means that they are pooling trials in which the go-cue has occurred and trials in which it has yet to appear. This seems problematic. Figure 5B doesn't do this, but it isn't clear what one is to take from this panel.

We apologize for not providing sufficient details on the control task and its related analyses. The reviewers are correct, during the *delayed-reach* task, a brief visual cue appeared on one side of the screen (left or right, randomly selected; 500 ms duration), and after a variable delay (1.1 – 3 s) monkeys made a reach movement towards that location.

For the mean activity shown in panels 5A and 5B we used all trials up the time at which the go-cue was presented. In this manner, every trial contributes to the mean at elapsed time 1.1 s. After that time point, there is a progressive “attrition” of trials so that the mean is calculated only for the trials whose delay duration was larger than the time indicated by the elapsed time axis. In other words, the mean is calculated with the available trials up to any given point in time, without the go-cue activity. The effect of this attrition can be seen in the increasing standard error of the mean activity in panel 5C. This averaging method allows using all available information up to a given point in time. We now explain this, in a new Materials and methods section:

“Delayed-reach control task: In this task monkeys were required wait a variable delay period before making a reach movement to one side of the screen signaled by a brief visual stimulus (Figure 5). […] Then, there is a progressive attrition of trials so that for the 3 s time point ~300 trials contribute to the mean. Figure 5B shows every trial in which the visual cue appeared on the left.”